# HYPER-REGULARIZATION: A FRAMEWORK OF ADAPTIVE CHOICE FOR THE LEARNING RATE IN GRADIENT DESCENT

## ABSTRACT

We present a novel approach for adaptively selecting the learning rate in gradient descent methods. Specifically, we impose a regularization term on the learning rate via a generalized distance, and cast the joint updating process of the parameter and the learning rate into a maxmin problem. Some existing schemes such as AdaGrad (diagonal version) and WNGrad can be rederived from our approach. Based on our approach, the updating rules for the learning rate do not rely on the smoothness constant of optimization problems and are robust to the initial learning rate. We theoretically analyze our approach in full batch and online learning settings, which achieves comparable performances with other first-order gradient-based algorithms in terms of accuracy as well as convergence rate.

## 1 INTRODUCTION

The automatic choice of the learning rate remains crucial in improving the efficiency of gradient descent algorithms, especially for solving nonconvex optimization problems. It is desirable to adaptively update the learning rate during the training process with a certain strategy. The convergence guarantees of such a strategy in theories usually require the Lipschitz constant or smoothness constant of the objective function to be explicitly known (Nesterov, 2013; Bubeck et al., 2015), which is inaccessible in most cases, e.g., in deep neural networks.

Using the received gradient information to adjust the current learning rate is a natural approach. In particular, the Steepest Descent uses the received gradient direction and an exact or inexact line search to obtain proper learning rates. Another important idea is to approximate second-order methods like Newton method (Nocedal & Wright, 2006) and Quasi-Newton Methods (Liu & Nocedal, 1989). Along this idea, for example, the Barzilai-Borweinin (BB) method (Barzilai & Borwein, 1988) in classical optimization and AdaGrad (Duchi et al., 2011) in online learning have been proposed.

In this paper we propose a novel framework to learn the learning rate that we call *Hyper-Regularization*. More specifically, we regard the learning rate as a hyperparameter and cast its adaptive choice with the parameter training into a joint process. We formulate this process as a maxmin framework by imposing a regularizer on the hyperparameter. Furthermore, we demonstrate that AdaGrad and WNGrad (Wu et al., 2018) can be derived using a streamlined scheme based on Hyper-Regularization.

In addition to solving the saddle point problem exactly, we also provide an alternating strategy to solve the problem approximately. We respectively give theoretical analysis for these two updating rules in full batch setting and online learning setting. Specifically, our results of runtime bounds in full batch setting and regret bounds in online learning setting are comparable to the best known bound in corresponding settings and indicate our algorithms converge for any initial learning rate.

### 1.1 RELATED WORK

Steepest Descent uses the received gradient direction and an exact or inexact line search to obtain proper learning rates. Although Steepest Descent uses the direction that descends most and the best learning rate that gives the most function reduction, Steepest Descent may converge very slow for

convex quadratic functions when the Hessian matrix is ill-conditioned (see, Yuan, 2008). In practice, some line search conditions such as Goldstein conditions or Wolfe conditions (see, Fletcher, 2013) can be applied to compute the learning rate. In online or stochastic settings, one observes stochastic gradients rather than exact gradients and line search methods become less effective.

The Barzilai-Borwein method (Barzilai & Borwein, 1988) which was motivated by quasi-Newton methods presents a surprising result that it could lead to superlinear convergence in convex quadratic problem of two variables. Although numerical results often show the Barzilai-Borwein method converges superlinearly in solving nonlinear optimization problem, no superlinear convergence results have been established even for an $n$-dimensional strictly convex quadratic problem with the order $n > 2$ (Barzilai & Borwein, 1988; Dai, 2013). In minimizing the sum of cost functions and stochastic setting, SGD-BB proposed by Tan et al. (2016) takes the average of the stochastic gradients in one epoch as an estimation of the full gradient. But this approach can not directly be applied to online learning settings.

In online convex optimization (Zinkevich, 2003; Shalev-Shwartz et al., 2012; Hazan et al., 2016), AdaGrad (Duchi et al., 2011) adapts the learning rate on per parameter basis dynamically. Intuitively, AdaGrad constructs approximation to the Hessian with diagonal of accumulated outer products of gradients. This leads to many variants such as RMSProp (Tieleman & Hinton, 2012), AdaDelta (Zeiler, 2012), Adam (Kingma & Ba, 2015), etc.

Additionally, Cruz (2011) analyzed Adaptive Stochastic Gradient Descent (ASGD) which is a generalization of Kesten's accelerated stochastic approximation algorithm (Kesten et al., 1958) for the high-dimensional case. ASGD uses a monotone decreasing function with respect to a time variable to get learning rates. Recently, Baydin et al. (2018) proposed Hyper-Gradient Descent to learn the global learning rate in SGD, SGD with Nesterov momentum and Adam. Hyper-Gradient Descent can be viewed as an approximate line search method in the online learning setting and it uses the update rule for the previous step to optimize the leaning rate in the current step. However, Hyper-Gradient Descent has no theoretical guarantee.

It is worth mentioning that Gupta et al. (2017) proposed a framework named Unified Adaptive Regularization from which AdaGrad and Online Newton Step (Hazan et al., 2007) can be derived. However, Unified Adaptive Regularization gives an approach for approximating the Hessian matrix in second order methods.

## 1.2 NOTATION

Before introducing our approach, we present the notation. We denote the set $\{x > 0 : x \in \mathbb{R}\}$ by $\mathbb{R}_{++}$. For two vectors $\boldsymbol{a}, \boldsymbol{b} \in \mathbb{R}^d$, we use $\boldsymbol{a}/\boldsymbol{b}$ to denote element-wise division, $\boldsymbol{a} \circ \boldsymbol{b}$ for element-wise product (the symbol $\circ$ will be omitted in the explicit context), $\boldsymbol{a}^n = (a_1^n, a_2^n, \ldots, a_d^n)$, and $\boldsymbol{a} \geq \boldsymbol{b}$ if $a_j \geq b_j$ for all $j$. Let $\mathbf{1}$ be the vector of ones with an appropriate size, and $\mathrm{diag}(\boldsymbol{\beta})$ be a diagonal matrix with the elements of the vector $\boldsymbol{\beta}$ on the main diagonal. In addition, we define $\|\boldsymbol{a}\|_A = \sqrt{\langle \boldsymbol{a}, A\boldsymbol{a} \rangle}$.

Given a set $\mathcal{X} \subseteq \mathbb{R}^d$, a function $f : \mathcal{X} \to \mathbb{R}$ is said to satisfy $f \in C_L^{1,1}(\mathcal{X})$ if $f$ is continuously differentiable on $\mathcal{X}$, and the derivative of $f$ is Lipschitz continuous on $\mathcal{X}$ with constant $L$:

$$\|\nabla f(\boldsymbol{x}) - \nabla f(\boldsymbol{y})\|_2 \leq L\|\boldsymbol{x} - \boldsymbol{y}\|_2.$$

More general definition can be found in Nesterov (2013).

## 1.3 PROBLEM STATEMENT

For an online learning problem, a learner faces a sequence of convex functions $\{f_t\}$ with the same domain $\mathcal{X} \subseteq \mathbb{R}^d$, receives (sub)gradient information $\boldsymbol{g}_t \in \partial f_t(\boldsymbol{x}_t)$ at each step $t$, and predicts a point $\boldsymbol{x}_{t+1} \in \mathcal{X}$.

Our theoretical analysis is based on two settings: full batch setting and online learning setting. A full batch setting (or optimization setting) is to optimize a certain function $F$ with exact gradient at each step, i.e., $f_t = F$. In this setting we assume $F \in C_L^{1,1}$ but it is not necessarily convex. We analyze the convergence of our algorithms by capping the runtime $T$ such that the minimum value

of the norm of received gradients so far is less than a given error accuracy $\varepsilon$, that is,

$$\min_{t=0:T-1} \|\nabla F(\boldsymbol{x}_t)\|_2^2 \leq \varepsilon.$$

In online learning settings, our analysis follows from Duchi et al. (2011) and Kingma & Ba (2015). We only assume that $f_t$'s are convex and try to give an upper bound for the regret

$$R(T) = \sum_{t=0}^{T-1} f_t(\boldsymbol{x}_t) - \min_{\boldsymbol{x} \in \mathcal{X}} \sum_{t=0}^{T-1} f_t(\boldsymbol{x}). \tag{1}$$

## 2 HYPER-REGULARIZATION

Following the setting in AdaGrad (Duchi et al., 2011), we consider a generalization of the standard (sub)gradient descent as

$$\boldsymbol{x}_{t+1} = \Pi_{\mathcal{X}}^{\text{diag}(\boldsymbol{\beta}_t)^{1/2}} \left( \boldsymbol{x}_t - \text{diag}(\boldsymbol{\beta}_t)^{-1/2} \boldsymbol{g}_t \right) = \arg\min_{\boldsymbol{x} \in \mathcal{X}} \left\| \boldsymbol{x}_t - \text{diag}(\boldsymbol{\beta}_t)^{-1/2} \boldsymbol{g}_t \right\|_{\text{diag}(\boldsymbol{\beta}_t)^{1/2}}^2. \tag{2}$$

This procedure can be viewed as the minimization problem:

$$\min_{\boldsymbol{x} \in \mathcal{X}} \langle \boldsymbol{g}_t, \boldsymbol{x} - \boldsymbol{x}_t \rangle + \frac{1}{2} \|\boldsymbol{x} - \boldsymbol{x}_t\|_{\text{diag}(\boldsymbol{\beta}_t)}^2. \tag{3}$$

To derive our hyper-regularization approach, we then formulate this minimization problem as a saddle point problem by adding a hyper-regularizer about the difference between the new learning rate $\boldsymbol{\beta}$ and an auxiliary vector $\boldsymbol{\eta}_t$. Accordingly, we have

$$\max_{\boldsymbol{\beta} \in \mathcal{B}_t} \min_{\boldsymbol{x} \in \mathcal{X}} \Psi_t(\boldsymbol{x}, \boldsymbol{\beta}) \triangleq \langle \boldsymbol{g}_t, \boldsymbol{x} - \boldsymbol{x}_t \rangle + \frac{1}{2} \Big( \|\boldsymbol{x} - \boldsymbol{x}_t\|_{\text{diag}(\boldsymbol{\beta})}^2 - D(\boldsymbol{\beta}, \boldsymbol{\eta}_t) \Big), \tag{4}$$

where $D(\boldsymbol{\beta}, \boldsymbol{\eta})$ is defined as our hyper-regularizer and $\mathcal{B}_t$ is a subset in $\mathbb{R}^d$. We solve the saddle point problem for new predictor and new learning rate.

Our framework stems from the work of Daubechies et al. (2010), where the authors adjust the weights of the weighted least squares problem by solving an extra objective function which added a regularizer about the weights to origin objective function. The following subsections will explain some details about our framework.

### 2.1 THE $\varphi$-DIVERGENCE

It is reasonable to choose a distance function to measure the difference between $\boldsymbol{\beta}$ and $\boldsymbol{\eta}$. In this paper, we use the $\varphi$-divergence[1] as our hyper-regularizer.

**Definition 1** ($\varphi$-divergence). *Let $\varphi$: $\mathbb{R}_{++} \to \mathbb{R}$ be a differentiable strongly convex function in $\mathbb{R}_{++}$ such that $\varphi(1) = \varphi'(1) = 0$, where $\varphi'$ is the derivative function of $\varphi$. For such a function $\varphi$, the function $D_\varphi$: $\mathbb{R}_{++}^d \times \mathbb{R}_{++}^d \to \mathbb{R}$, which is define by*

$$D_\varphi(\boldsymbol{u}, \boldsymbol{v}) \triangleq \sum_{j=1}^d v_j \varphi(u_j/v_j),$$

*is referred to as the $\varphi$-divergence.*

**Remark.** *Note that convex function $\varphi$ with $\varphi(1) = \varphi'(1) = 0$ satisfies $\varphi(z) \geq 0$ for all $z > 0$, thus $D_\varphi(\boldsymbol{u}, \boldsymbol{v}) \geq 0$ for all $\boldsymbol{u}, \boldsymbol{v} \in \mathbb{R}_{++}^d$, with equality iff $\boldsymbol{u} = \boldsymbol{v}$.*

**Remark.** *For any convex function $f$, $\varphi(z) = f(z) - f'(1)(z-1) - f(1)$ is a proper function for our $\varphi$-divergence.*

**Remark.** *In our framework, in order to solve the problem (4) feasibly, we always assume that $\lim_{z \to +\infty} \varphi'(z) = +\infty$.*

---

[1]Comparing with Bregman divergence, $\varphi$-divergence requires nonnegative $\boldsymbol{\beta}$ and $\boldsymbol{\eta}_t$ which is apparent for learning rates.

If one only requires $\varphi$ to be convex and $\varphi(1) = 0$, the resulting distance function $D_\varphi$ is called a $f$-divergence (Liese & Vajda, 1987; 2006). The $f$-divergence has been widely applied in statistical machine learning (e.g., Nguyen et al., 2009).

Using the $\varphi$-divergence as our hyper-regularizer, we can rewrite the problem (4) as

$$\max_{\boldsymbol{\beta} \in \mathcal{B}_t} \min_{\boldsymbol{x} \in \mathcal{X}} \Psi_t(\boldsymbol{x}, \boldsymbol{\beta}) \triangleq \boldsymbol{g}_t^\top (\boldsymbol{x} - \boldsymbol{x}_t) + \frac{1}{2} \|\boldsymbol{x} - \boldsymbol{x}_t\|_{\mathrm{diag}(\boldsymbol{\beta})}^2 - \frac{1}{2} D_\varphi(\boldsymbol{\beta}, \boldsymbol{\eta}_t), \text{ or} \tag{5}$$

$$\max_{\boldsymbol{\beta} \in \mathcal{B}_t} \min_{\boldsymbol{x} \in \mathcal{X}} \Psi_t(\boldsymbol{x}, \boldsymbol{\beta}) \triangleq \sum_{j=1}^d g_{t,j}(x_j - x_{t,j}) + \frac{1}{2} \left( \beta_j (x_j - x_{t,j})^2 - \eta_{t,j} \varphi(\beta_j / \eta_{t,j}) \right). \tag{6}$$

The form of problem (6) implies that we can solve the problem for each dimension separately, and only a few extra calculations are required for each step.

## 2.2 MAX-MIN OR MIN-MAX

Consider a problem similar to (5),

$$\min_{\boldsymbol{x} \in \mathcal{X}} \max_{\boldsymbol{\beta} \in \mathcal{B}_t} \Psi_t(\boldsymbol{x}, \boldsymbol{\beta}). \tag{7}$$

The solution of problem (5) is same as (7) in unconstrained case[2], i.e. $\mathcal{X} = \mathbb{R}^d, \mathcal{B}_t = \mathbb{R}_{++}^d$. However, if we set $\mathcal{B}_t = [b_{t,1}, B_{t,1}] \times \cdots \times [b_{t,d}, B_{t,d}]$ to constrain the range of $\boldsymbol{\beta}_{t+1}$ and suppose $\mathcal{X} = \mathbb{R}^d$, the solution of (7) is more difficult to get from the solution of the unconstrained problem, while the solution of (5) can be easily obtained by clipping the solution of the unconstrained problem to required range (see Lemma 2). Thus, we choose (5) as our basic problem.

**Lemma 2.** *Suppose that* $\mathcal{B}_t = [b_{t,1}, B_{t,1}] \times \cdots \times [b_{t,d}, B_{t,d}]$, *and* $\mathcal{X} = \mathbb{R}^d$. *Let* $\boldsymbol{\beta}^*$ *be the solution of unconstrained problem* $\max_{\boldsymbol{\beta}}(\min_{\boldsymbol{x}} \Psi_t(\boldsymbol{x}, \boldsymbol{\beta}))$. *Then the solution of problem* $\max_{\boldsymbol{\beta} \in \mathcal{B}_t}(\min_{\boldsymbol{x}} \Psi_t(\boldsymbol{x}, \boldsymbol{\beta}))$ *is*

$$\beta_j = \min\{\max\{\beta_j^*, b_{t,j}\}, B_{t,j}\}, \text{ for } j = 1, \cdots, d.$$

## 2.3 SELECTION OF $\boldsymbol{\eta}_t$

There are many ways to choose the sequence $\{\boldsymbol{\eta}_t\}$:

- The sequence $\{\boldsymbol{\eta}_t\}$ is chosen in advance, before our methods start the job. For example, set 1) $\boldsymbol{\eta}_t = \eta \mathbf{1}$; 2) $\boldsymbol{\eta}_t = \eta \sqrt{t+1} \mathbf{1}$, where $\eta > 0$ is a prespecified constant.
- $\boldsymbol{\eta}_t$ can be obtained from other adaptive learning rate methods such as AdaGrad.
- Set $\boldsymbol{\eta}_t = \boldsymbol{\beta}_t$. In this case, the hyper-regularizer is the penalty for the change between $\boldsymbol{\beta}_{t+1}$ and $\boldsymbol{\beta}_t$.

The first two ways can be treated as a smoothing technique to stabilize the learning rate. We want to ensure the learning rate sequence $\{\boldsymbol{\beta}_t\}$ close to another learning rate sequence $\{\boldsymbol{\eta}_t\}$. Although setting $\boldsymbol{\eta}_t = \boldsymbol{\beta}_t$ is our main focus, we keep the sequence of $\{\boldsymbol{\eta}_t\}$ to maintain this flexibility.

## 2.4 ISOTROPIC HYPER-REGULARIZATION

We now show a special form of our framework that only adaptively maintains a single scalar learning rate. We modify Hyper-regularization so that $\boldsymbol{\beta}$ is optimized over the set $\mathcal{B}_t \subseteq \{\theta \mathbf{1} : \theta \in \mathbb{R}_{++}\}$ of all positive multiples of the vector $\mathbf{1} \in \mathbb{R}^d$. Let $\boldsymbol{\eta}_t = \eta_t \mathbf{1}$, and rewrite problem (5) as

$$\max_{\beta \mathbf{1} \in \mathcal{B}_t} \min_{\boldsymbol{x} \in \mathcal{X}} \Psi_t(\boldsymbol{x}, \beta) = \boldsymbol{g}_t^\top (\boldsymbol{x} - \boldsymbol{x}_t) + \frac{\beta}{2} \|\boldsymbol{x} - \boldsymbol{x}_t\|_2^2 - \frac{1}{2} \eta_t \varphi(\beta / \eta_t). \tag{8}$$

We prefer to use Isotropic Hyper-Regularization as smoothing technique in online learning settings.

---

[2]Using partial derivative, we can obtain the saddle point of $\Psi_t$ and ensure this fact.

## 3 UPDATE RULES AND ALGORITHMS

In this section we present two update rules of our hyper-regularization framework. The first update rule is solving the saddle point problem (5) exactly. That is,

$$\Psi_t(\boldsymbol{x}_{t+1}, \boldsymbol{\beta}_{t+1}) = \max_{\boldsymbol{\beta} \in \mathcal{B}_t} \min_{\boldsymbol{x} \in \mathcal{X}} \Psi_t(\boldsymbol{x}, \boldsymbol{\beta}). \tag{9}$$

The following lemma and Algorithm 1 give the concrete scheme of our iterations.

**Lemma 3.** *Considering problem (6) without constraints and solving the problem exactly, we get new predictor $\boldsymbol{x}_{t+1}$ and new learning rate $\boldsymbol{\beta}_{t+1}$ such that*

$$\beta_{t+1,j}^2 \varphi'(\beta_{t+1,j}/\eta_{t,j}) = g_{t,j}^2, j = 1, \ldots, d, \tag{10}$$

$$\boldsymbol{x}_{t+1} = \boldsymbol{x}_t - \boldsymbol{g}_t/\boldsymbol{\beta}_{t+1}.$$

---

**Algorithm 1** GD with Hyper-regularization

---

**Input:** $\boldsymbol{\beta}_0 > 0, \boldsymbol{x}_0$
1: **for** $t = 1$ to $T$ **do**
2:    Suffer loss $f_t(\boldsymbol{x}_t)$;
3:    Receive subgradient $\boldsymbol{g}_t \in \partial f_t(\boldsymbol{x}_t)$ of $f_t$ at $\boldsymbol{x}_t$;
4:    Update $\beta_{t+1,j}$ as the solution of the equation $\beta^2 \varphi'(\beta/\eta_{t,j}) = g_{t,j}^2, j = 1, \ldots, d$;
5:    Update $\boldsymbol{x}_{t+1} = \boldsymbol{x}_t - \boldsymbol{g}_t/\boldsymbol{\beta}_{t+1}$;
6: **end for**

---

**Remark.** *Note that AdaGrad (Duchi et al., 2011) and WNGrad (Wu et al., 2018) are special cases of Algorithm 1 with a particular choice of $\varphi$ (detailed derivation in Appendix B).*

- *If $\varphi(z) = z + \frac{1}{z} - 2$, then we can derive AdaGrad from Algorithm 1.*

- *If $\varphi(z) = \frac{1}{z} - \log(\frac{1}{z}) - 1$, then we can derive WNGrad from Algorithm 1.*

### 3.1 ALTERNATING UPDATE RULE

However, it is sometimes difficult to solve the equation (10), especially in the case where $\varphi$ is a quadratic function (equation (10) will be a cubic equation in one variable for any $j$). In practice, using an alternating update strategy is more recommended. Under the assumption that the optimal value of $\boldsymbol{\beta}$ is close to $\boldsymbol{\eta}_t$, we solve an approximate equation for $\boldsymbol{\beta}_{t+1}$

$$\boldsymbol{\beta}_{t+1} = \arg\max_{\boldsymbol{\beta} \in \mathcal{B}_t} \Psi_t \left( \arg\min_{\boldsymbol{x} \in \mathcal{X}} \Psi_t(\boldsymbol{x}, \boldsymbol{\eta}_t), \boldsymbol{\beta} \right), \tag{11}$$

and update the new predictor $\boldsymbol{x}_{t+1}$ using

$$\boldsymbol{x}_{t+1} = \arg\min_{\boldsymbol{x} \in \mathcal{X}} \Psi_t(\boldsymbol{x}, \boldsymbol{\beta}_{t+1}).$$

Applying the alternating update rule under the same assumption in Lemma 3, we obtain the following lemma and Algorithm 2.

**Lemma 4.** *Considering problem (6) without constraint and following from the alternating update rule, we get new predictor $\boldsymbol{x}_{t+1}$ and new learning rate $\beta_{t+1}$ as*

$$\beta_{t+1,j} = \eta_{t,j}(\varphi')^{-1}(g_{t,j}^2/\eta_{t,j}^2), j = 1, \ldots, d, \tag{12}$$

$$\boldsymbol{x}_{t+1} = \boldsymbol{x}_t - \boldsymbol{g}_t/\boldsymbol{\beta}_{t+1}.$$

---

**Algorithm 2** GD with Hyper-regularization using alternating update rule

---

**Input:** $\boldsymbol{\beta}_0 > 0, \boldsymbol{x}_0$
1: **for** $t = 1$ to $T$ **do**
2:    Suffer loss $f_t(\boldsymbol{x}_t)$;
3:    Receive $\boldsymbol{g}_t \in \partial f_t(\boldsymbol{x}_t)$ of $f_t$ at $\boldsymbol{x}_t$;
4:    Update $\beta_{t+1,j} = \eta_{t,j}(\varphi')^{-1}(g_{t,j}^2/\eta_{t,j}^2), j = 1, \ldots, d$;
5:    Update $\boldsymbol{x}_{t+1} = \boldsymbol{x}_t - \boldsymbol{g}_t/\boldsymbol{\beta}_{t+1}$;
6: **end for**

---

Computing the inverse function of $\varphi'$ is usually easier than solving the equation (10) in practice, especially for the widely used $\varphi$-divergences (more details can be found in Appendix D.1).

**Remark.** *If $\boldsymbol{\eta}_t = \boldsymbol{\beta}_t$, the following simplified alternating rule can be employed:*

$$\boldsymbol{x}_{t+1} = \arg\min_{\boldsymbol{x} \in \mathcal{X}} \Psi_t(\boldsymbol{x}, \boldsymbol{\beta}_t),$$

$$\boldsymbol{\beta}_{t+1} = \arg\max_{\boldsymbol{\beta} \in \mathcal{B}_t} \Psi_t(\boldsymbol{x}_{t+1}, \boldsymbol{\beta}).$$

*We leave the corresponding algorithm 3 in Appendix C.*

## 3.2 MONOTONICITY

Before giving more analysis, let us show monotonicity of both the two update rules. The monotonicity implies that only the property of convex function $\varphi$ on interval $[1, +\infty)$ will influence the efficiency of our algorithms.

**Lemma 5.** *$\boldsymbol{\beta}_{t+1}$ obtained from equation (9) or (11) satisfies $\boldsymbol{\beta}_{t+1} \geq \boldsymbol{\eta}_t$.*

When setting $\boldsymbol{\beta}_t = \boldsymbol{\eta}_t$, we have that $\boldsymbol{\beta}_{t+1} \geq \boldsymbol{\beta}_t$.

## 4 THEORETICAL ANALYSIS

In this section we always set $\boldsymbol{\eta}_t = \boldsymbol{\beta}_t$ and assume that $\boldsymbol{x}$ and $\boldsymbol{\beta}$ are unconstrained, i.e., $\mathcal{X} = \mathbb{R}^d$ and $\mathcal{B}_t = \mathbb{R}_{++}^d$. We first discuss the convergence rate of the two update rules in full batch setting with assumption that the objective function $F$ is $L$-smooth but not necessarily convex in Section 4.1. Next we turn to online convex learning setting and establish a theorem about the regret bounds in Section 4.2. Our results for both the settings show that our algorithms are robust to the choice of initial learning rates and do not rely on the Lipschitz constant or smoothness constant.

### 4.1 ISOTROPIC HYPER-REGULARIZATION IN FULL BATCH SETTING

Recall that we set $f_t = F$ in the full batch setting, and assume that $F \in C_L^{1,1}$ without convexity. In this case, two update rules can be written as

$$\begin{cases} \beta_{t+1}^2 \varphi'(\beta_{t+1}/\beta_t) = \|\boldsymbol{g}_t\|_2^2, \\ \boldsymbol{x}_{t+1} = \boldsymbol{x}_t - \frac{1}{\beta_{t+1}} \boldsymbol{g}_t. \end{cases} \quad (13) \qquad \begin{cases} \beta_{t+1} = \beta_t(\varphi')^{-1}(\|\boldsymbol{g}_t\|_2^2/\beta_t^2), \\ \boldsymbol{x}_{t+1} = \boldsymbol{x}_t - \frac{1}{\beta_{t+1}} \boldsymbol{g}_t. \end{cases} \quad (14)$$

Next we show that both update rules (13) and (14) are robust to the choice of initial learning rate.

**Theorem 6.** *Suppose that $\varphi \in C_l^{1,1}([1, +\infty))$, $\varphi$ is $\alpha$-strongly convex, $F \in C_L^{1,1}(\mathbb{R}^d)$, and $F^* = \inf_{\boldsymbol{x}} F(\boldsymbol{x}) > -\infty$. For any $\varepsilon \in (0, 1)$, the sequence $\{\boldsymbol{x}_t\}$ obtained from update rules (13) or (14) satisfies*

$$\min_{j=0:T-1} \|\nabla F(\boldsymbol{x}_j)\|_2^2 \leq \varepsilon,$$

*after $T = \mathcal{O}\left(\frac{1}{\varepsilon}\right)$ steps.*

More detailed results of Theorem 6 for runtime can be found in Theorems 21 and 22 in Appendix I. Theorem 6 shows that both runtime of the two update rules can be bound as $\mathcal{O}(1/\varepsilon)$ for any constant $L$ and initial learning rate $\beta_0$. As a comparison, in classical convergence result ((1.2.13) in Nesterov (2013) or Theorem 20 in Appendix), the upper bound of runtime is $\mathcal{O}(1/\varepsilon)$ only for a certain range (related to $L$) of initial learning rates.

### 4.2 HYPER-REGULARIZATION IN ONLINE LEARNING SETTING

We now establish the result of convergence rate for Algorithms 1 and 2 in online convex learning, i.e., the $f_t$ are convex. Especially, we try to bound regrets (1) by $\mathcal{O}(\sqrt{T})$ for Algorithms 1 and 2.

**Theorem 7.** *Suppose that $\varphi \in C_l^{1,1}([1, +\infty))$, and $\varphi$ is $\alpha$-strongly convex. Assume that $\|g_t\|_\infty \leq G$, and $\|x_t - x^*\|_\infty \leq D_\infty$. Then the sequence $\{x_t\}$ obtained from Algorithm 1 satisfies*

$$2R(T) \leq \frac{(\alpha + D_\infty^2)\sqrt{2l\beta_0^2 + 4G^2}}{\alpha\beta_0} \sum_{j=1}^{d} \|g_{0:T-1,j}\|_2 + \beta_0\|x_0 - x^*\|_2^2,$$

*and the sequence $\{x_t\}$ obtained from Algorithm 2 (or 3) satisfies*

$$2R(T) \leq \left(1 + \frac{D_\infty^2}{\alpha}\right)\max\left\{\sqrt{2l}, \frac{2G}{\beta_0}\right\} \sum_{j=1}^{d} \|g_{0:T-1,j}\|_2 + \beta_0\|x_0 - x^*\|_2^2.$$

Note that under the assumption in Theorem 7, $\sum_{j=1}^{d} \|g_{0:T-1,j}\|_2 \leq dG\sqrt{T}$, hence $R(T) = \mathcal{O}(\sqrt{T})$. Our result is comparable to the best known bound for convex online learning problem (see Hazan et al., 2016; Duchi et al., 2011; Kingma & Ba, 2015).

## 5 EXPERIMENTS

In this paper our principal focus has been to develop a new approach for adaptive choice of the learning rate in first-order gradient-type methods. However, this new approach also brings some insights into the resulting algorithms. Thus, it is interesting to conduct empirical analysis of the learning algorithms with different choice methods for the learning rate.

### 5.1 THE SET-UP

To derive a learning algorithm from the Hyper-Regularization framework, we have to first give the $\varphi$ divergences (Pardo, 2005). Specifically, the algorithms from our framework in the following experiments are derived from the following four $\varphi$ divergences (full implementations are displayed in the Appendix D.1):

- $\varphi(t) = t\log t - t + 1$ from KL-divergence inducing KL algorithm.
- $\varphi(t) = -\log t + t - 1$ from Reverse KL-divergence inducing RKL algorithm.
- $\varphi(t) = (\sqrt{t} - 1)^2$ from Hellinger distance inducing Hellinger algorithm.
- $\varphi(t) = (t - 1)^2$ from $\chi^2$ distance inducing $\chi^2$ algorithm.

As mentioned in Section 3, even with a fixed $\varphi$ divergence, the generated algorithm still varies with different update rules. For simplicity, different update rules were compared in advance to select the specific one for any $\varphi$ divergence in the following experiments.

To maintain stable performance, the technique of growth clipping is applied to all algorithms in our framework. Actually, growth clipping fulfills the constraints placed on the increasing speed of $\beta_t$ by $\mathcal{B}_t$ in Lemma 2. Specifically, $\beta_{t+1}$ in our experiments falls in $[\beta_t, 2\beta_t]$. Detailed observations on how the $\beta_t$ of our algorithms increases are left in Appendix D.3.

Experiments involve the four algorithms generated above as well as other first-order gradient-based algorithms including SGD (with no learning rate decay), SGD-BB, and Hyper-Gradient Descent algorithms. These algorithms are evaluated on tasks of image classification with a logistic classifier on the databases of MNIST (LeCun et al., 2010) and CIFAR-10 (Krizhevsky & Hinton, 2009). Initial learning rate (in the usual sense, i.e., $1/\beta_0$) varies from $10^{-3}$ to $10^1$ for the test of convergence performance of these algorithms. Experiments are run using Tensorflow (Abadi et al., 2016), on a machine with Intel Xeon E5-2680 v4 CPU, 128 GB RAM, and NVIDIA Titan Xp GPU.

### 5.2 UPDATE RULE SELECTION

Taking the RKL algorithm as an example, we refer to algorithms deduced from Algorithm 1, 2, and 3 as $RKL_1$, $RKL_2$, and $RKL_3$. We train a two-layer neural network with a hidden layer of 500 units on the MNIST database. Experiments are in online learning setting with a batch size of 128, and $\ell_2$ regularization is applied with a coefficient of $10^{-4}$.

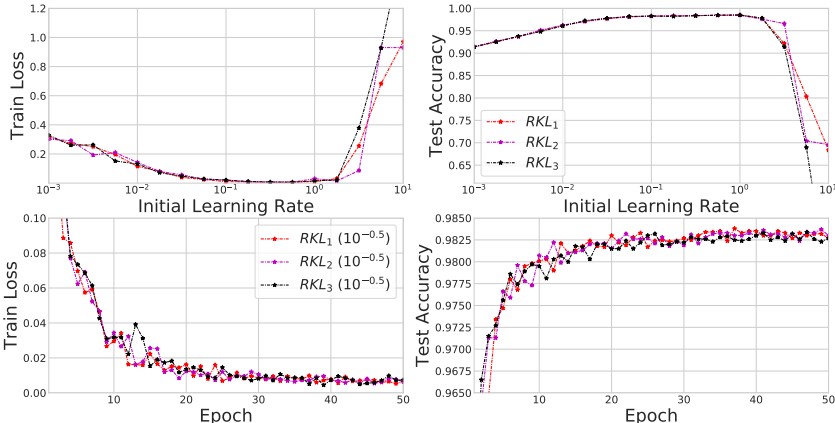

Figure 1: Convergence performances of RKL$_1$, RKL$_2$, and RKL$_3$ on the database of MNIST in online learning setting, *up:* performances at different initial learning rates, and *down:* the whole training process with the given initial learning rate in the bracket for each algorithm.

Figure 1 demonstrates the training loss and test accuracy of these algorithms with various initial learning rates. After fixing the learning rate with the least training loss, we compare their performances throughout the training process. Sharing comparable performances at small initial learning rate with RKL$_3$, RKL$_1$ and RKL$_2$ perform better at relatively large learning rate.

Generally speaking, Algorithm 1 often suffers a higher computation complexity than Algorithm 2 for the difficulty of getting an analytical solution. Detailed observations are left in Appendix D.1. Therefore, we apply the second update rule to our algorithms without explicit notifications.

## 5.3 FULL BATCH SETTING AND ONLINE LEARNING SETTING

We investigate our algorithms in the full batch setting on the MNIST database where algorithms receive the exact gradients of the objective loss function each iteration. [3]

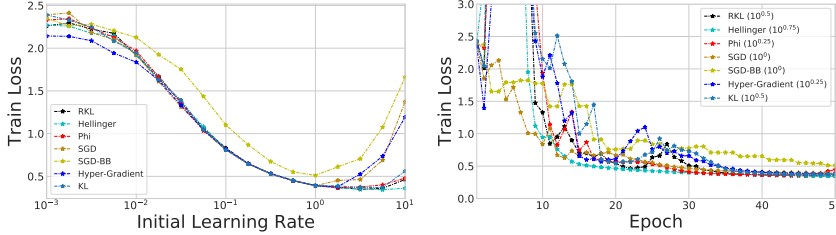

Figure 2: Convergence performances of algorithms on MNIST in the full batch setting. *left:* the train loss of the last training epoch at different initial learning rates, *right:* the whole training process with the given initial learning rate in the bracket for each algorithm.

In terms of online learning setting, we train a VGG Net (Simonyan & Zisserman, 2014) with batch normalization on the CIFAR-10 database with a batch size of 128, and an $\ell_2$ regularization coefficient of $10^{-4}$. We as well perform data augmentation as He et al. (2016) to improve the training.

Figure 2 and Figure 3 show the convergence performances on both settings, respectively. Achieving general comparable performances with other first-order gradient-based algorithms, our algorithms outperform most of other algorithms at risky large initial learning rate. Even at the fixed initial learning rate with the least training loss, the performances of our algorithms still achieve the same training performances with others.

---

[3]Since it is a pure optimization problem, testing performance is out of our main consideration.

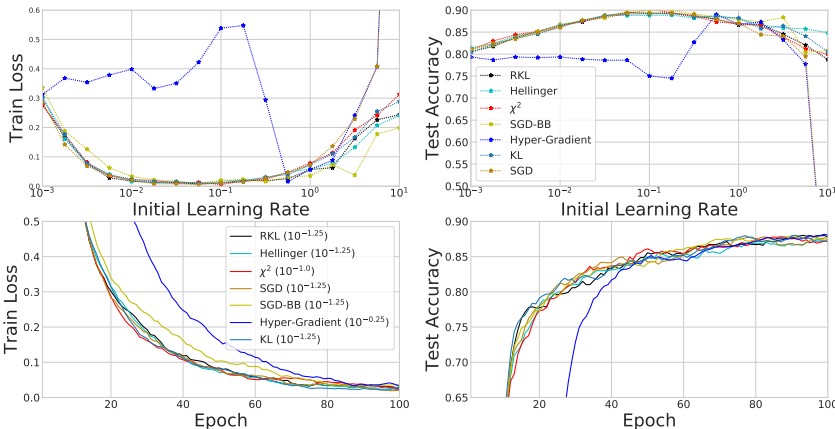

Figure 3: Convergence performances of algorithms on CIFAR-10 in the online learning setting, *left:* the train loss of the last training epoch at different initial learning rates, *right:* the whole training process with the given initial learning rate in the bracket for each algorithm.

## 6 DISCUSSION

As a supplement, we point out that logarithmic regret bounds under assumption $f_t$ is strongly convex, like Hazan et al. (2007); Mukkamala & Hein (2017), can be established with a different class of distance function. We leave the details in Appendix H.

In this paper, we propose a novel framework distinct from previous main approaches like line search and approximate second-order methods. It worth noting that Hyper-Regularization can generates efficient algorithms for optimization problems with regularization terms shown above and we expect new ideas of more efficient terms in the future.

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

## A    SOLUTION EXISTENCE

Note that the function $h(\beta) = \beta^2 \varphi'(\beta/\eta_{t,j})$ is an increasing continuous function and $\lim_{z \to +\infty} \varphi'(z) = +\infty$ $\varphi'(1) = 0$, so $[0, +\infty)$ is a subset of the range of $h(\beta)$ and the solution of (10) exists.
For the same reason, the solution of (12) exists.

## B    SPECIAL CASES OF ALGORITHM 1

In this section, We will point out that Adagrad (Duchi et al., 2011) and WNGrad (Wu et al., 2018) are special cases of Algorithm 1.
If we set $\varphi(z) = z + \frac{1}{z} - 2$, then the new learning rate $\boldsymbol{\beta}_{t+1}$ can be obtained by

$$\beta_{t+1,j}^2 \left( 1 - \frac{\beta_{t,j}^2}{\beta_{t+1,j}^2} \right) = g_{t,j}^2, \; j = 1, \cdots, d,$$

that implies,

$$\boldsymbol{\beta}_{t+1}^2 = \boldsymbol{\beta}_t^2 + \boldsymbol{g}_t^2,$$

and we drive AdaGrad from Hyper-Regularization.

Similarly, we can get WNGrad by setting $\varphi(z) = \frac{1}{z} - \log \frac{1}{z} - 1$. In fact, $\boldsymbol{\beta}_{t+1}$ employs update

$$\beta_{t+1,j}^2 \left( \frac{\beta_{t,j}(\beta_{t+1,j} - \beta_{t,j})}{\beta_{t+1,j}^2} \right) = g_{t,j}^2, \; j = 1, \cdots, d,$$

on the other words,

$$\boldsymbol{\beta}_{t+1} = \boldsymbol{\beta}_t + \frac{\boldsymbol{g}_t^2}{\boldsymbol{\beta}_t},$$

i.e., the update rule of WNGrad.

## C    SIMPLIFIED ALTERNATING UPDATE RULE

---

**Algorithm 3** GD with Hyper-regularization using simplified alternating update rule and $\eta_t = \boldsymbol{\beta}_t$

---

**Input:** $\boldsymbol{\beta}_0 > 0$, $\boldsymbol{x}_0$
1: **for** $t = 1$ to $T$ **do**
2:     Suffer loss $f_t(\boldsymbol{x}_t)$;
3:     Receive $\boldsymbol{g}_t \in \partial f_t(\boldsymbol{x}_t)$ of $f_t$ at $\boldsymbol{x}_t$;
4:     Update $\boldsymbol{x}_{t+1} = \boldsymbol{x}_t - \boldsymbol{g}_t / \boldsymbol{\beta}_t$;
5:     Update $\beta_{t+1,j} = \beta_{t,j}(\varphi')^{-1}(g_{t,j}^2/\beta_{t,j}^2)$, $j = 1, \ldots, d$;
6: **end for**

---

The weakness of the simplified alternating update rule is that we get the new predictor $\boldsymbol{x}_{t+1}$ by $\boldsymbol{\beta}_t$ which is unrelated to current gradient $\boldsymbol{g}_t$. The regret bound of Algorithm 3 has been shown in Theorem 7.

## D    SUPPLEMENTARY FOR NUMERICAL EXPERIMENTS

Full versions of Hellinger, KL, RKL, and $\chi^2$ algorithms through Algorithm 1, 2, and 3 are listed below. Apparently, it is of great complexity to compute an analytical solution for $\boldsymbol{\beta}_{t+1}$ for the equations marked red.

### D.1 FULL VERSIONS OF GENERATED ALGORITHMS

$$KL_1 \begin{cases} \boldsymbol{\beta}_{t+1}^2 \log(\boldsymbol{\beta}_{t+1}/\boldsymbol{\beta}_t) = \boldsymbol{g}_t^2 \\ \boldsymbol{x}_{t+1} = \boldsymbol{x}_t - \boldsymbol{g}_t/\boldsymbol{\beta}_{t+1} \end{cases}$$

$$KL_2 \begin{cases} \boldsymbol{\beta}_{t+1} = \boldsymbol{\beta}_t \exp(g_t^2/\boldsymbol{\beta}_t^2) \\ \boldsymbol{x}_{t+1} = \boldsymbol{x}_t - \boldsymbol{g}_t/\boldsymbol{\beta}_{t+1} \end{cases}$$

$$KL_3 \begin{cases} \boldsymbol{x}_{t+1} = \boldsymbol{x}_t - \boldsymbol{g}_t/\boldsymbol{\beta}_t \\ \boldsymbol{\beta}_{t+1} = \boldsymbol{\beta}_t \exp(g_t^2/\boldsymbol{\beta}_t^2) \end{cases}$$

$$RKL_1 \begin{cases} \boldsymbol{\beta}_{t+1} = \frac{1}{2}\left(\boldsymbol{\beta}_t + \sqrt{\boldsymbol{\beta}_t^2 + 4\boldsymbol{g}_t^2}\right) \\ \boldsymbol{x}_{t+1} = \boldsymbol{x}_t - \boldsymbol{g}_t/\boldsymbol{\beta}_{t+1} \end{cases}$$

$$RKL_2 \begin{cases} \boldsymbol{\beta}_{t+1} = \boldsymbol{\beta}_t^3/(\boldsymbol{\beta}_t^2 - \boldsymbol{g}_t^2) \\ \boldsymbol{x}_{t+1} = \boldsymbol{x}_t - \boldsymbol{g}_t/\boldsymbol{\beta}_{t+1} \end{cases}$$

$$RKL_3 \begin{cases} \boldsymbol{x}_{t+1} = \boldsymbol{x}_t - \boldsymbol{g}_t/\boldsymbol{\beta}_t \\ \boldsymbol{\beta}_{t+1} = \boldsymbol{\beta}_t^3/(\boldsymbol{\beta}_t^2 - \boldsymbol{g}_t^2) \end{cases}$$

$$H_1 \begin{cases} \boldsymbol{\beta}_{t+1}^2(1 - \sqrt{\boldsymbol{\beta}_t/\boldsymbol{\beta}_{t+1}}) = \boldsymbol{g}_t^2 \\ \boldsymbol{x}_{t+1} = \boldsymbol{x}_t - \boldsymbol{g}_t/\boldsymbol{\beta}_{t+1} \end{cases}$$

$$H_2 \begin{cases} \boldsymbol{\beta}_{t+1} = \boldsymbol{\beta}_t^5/(\boldsymbol{\beta}_t^2 - \boldsymbol{g}_t^2)^2 \\ \boldsymbol{x}_{t+1} = \boldsymbol{x}_t - \boldsymbol{g}_t/\boldsymbol{\beta}_{t+1} \end{cases}$$

$$H_3 \begin{cases} \boldsymbol{x}_{t+1} = \boldsymbol{x}_t - \boldsymbol{g}_t/\boldsymbol{\beta}_t \\ \boldsymbol{\beta}_{t+1} = \boldsymbol{\beta}_t^5/(\boldsymbol{\beta}_t^2 - \boldsymbol{g}_t^2)^2 \end{cases}$$

$$\chi_1^2 \begin{cases} 2\boldsymbol{\beta}_{t+1}^2(\boldsymbol{\beta}_{t+1}/\boldsymbol{\beta}t - 1) = \boldsymbol{g}_t^2 \\ \boldsymbol{x}_{t+1} = \boldsymbol{x}_t - \boldsymbol{g}_t/\boldsymbol{\beta}_{t+1} \end{cases}$$

$$\chi_2^2 \begin{cases} \boldsymbol{\beta}_{t+1} = \boldsymbol{\beta}_t\left(1 + \boldsymbol{g}_t^2/(2\boldsymbol{\beta}_t^2)\right) \\ \boldsymbol{x}_{t+1} = \boldsymbol{x}_t - \boldsymbol{g}_t/\boldsymbol{\beta}_{t+1} \end{cases}$$

$$\chi_3^2 \begin{cases} \boldsymbol{x}_{t+1} = \boldsymbol{x}_t - \boldsymbol{g}_t/\boldsymbol{\beta}_t \\ \boldsymbol{\beta}_{t+1} = \boldsymbol{\beta}_t\left(1 + \boldsymbol{g}_t^2/(2\boldsymbol{\beta}_t^2)\right) \end{cases}$$

### D.2 VERSION COMPARISON ON OTHER ALGORITHMS

Figure 4, 5 and 6 shows the results of comparison between versions of KL, Hellinger, and $\chi^2$ algorithms on the base of MNIST in the online learning setting.

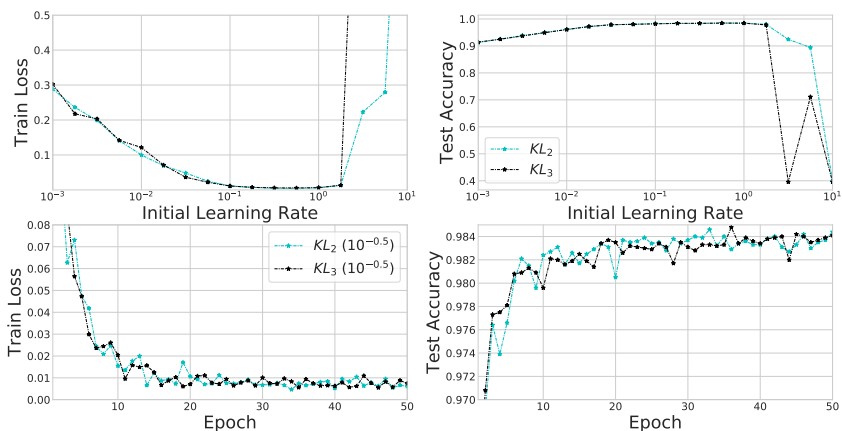

Figure 4: Convergence performances of $KL_2$ and $KL_3$ on the database of MNIST in online learning setting, *up:* at different initial learning rates, and *down:* the whole training process with the given initial learning rate in the bracket.

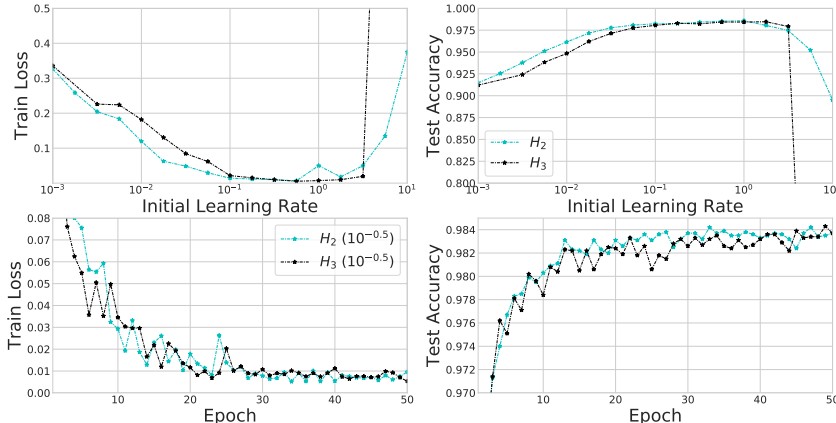

Figure 5: Convergence performances of $H_2$ and $H_3$ on the database of MNIST in online learning setting, *up:*at different initial learning rates, and *down:*the whole training process with the given initial learning rate in the bracket.

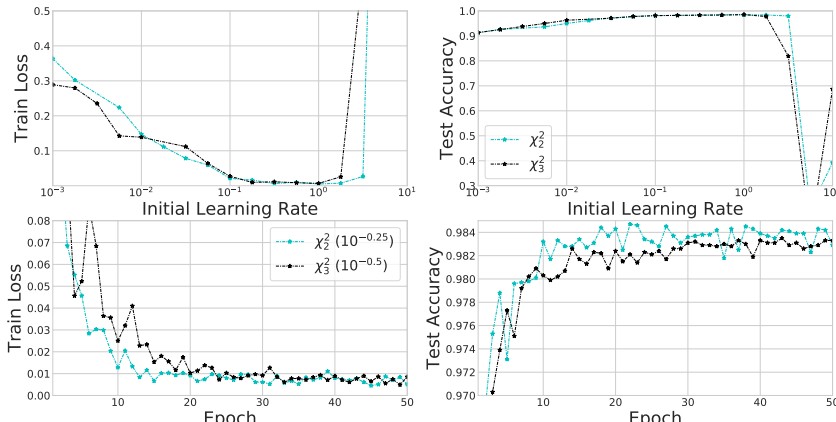

Figure 6: Convergence performances of $\chi_2^2$ and $\chi_3^2$ on the database of MNIST in online learning setting, *up:*at different initial learning rates, and *down:*the whole training process with the given initial learning rate in the bracket.

### D.3 FIGURES ON $\beta_t$ INCREASING PROCESS

All the following experiments are based on the MNIST database in online learning setting with a batch size of 128. For simplicity, all initial $\beta_0$s are fixed to be $10^{-0.5}$. Considering the sparse feature of parameters in this task, figure 7 only observe the changing process of the maximum of $\beta_t$.

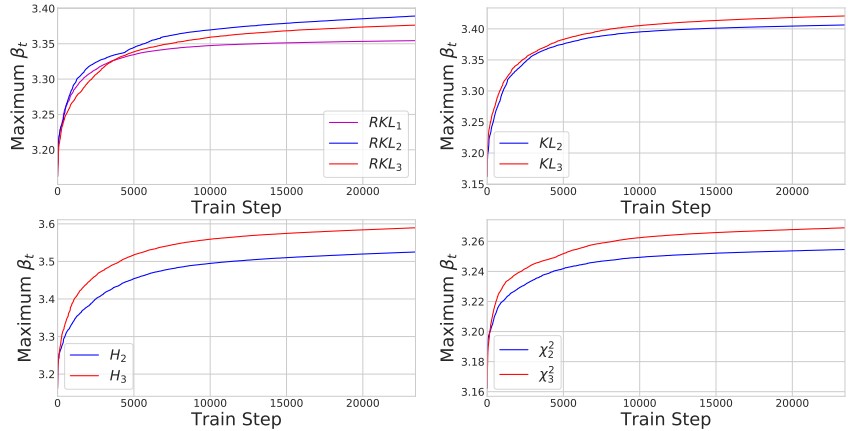

Figure 7: The values of the maximum $\boldsymbol{\beta}_t$ at each training step for various algorithms.

## E    PROOF OF LEMMA 2

*Proof.* First, it is trivial to get

$$\Psi_{t,\boldsymbol{x}}(\boldsymbol{\beta}) \triangleq min_{\boldsymbol{x}}\Psi_t(\boldsymbol{x},\boldsymbol{\beta}) = \Psi_t\left(\boldsymbol{x}_t - \frac{\boldsymbol{g}_t}{\boldsymbol{\beta}},\boldsymbol{\beta}\right)$$

$$= -\frac{1}{2}\|\boldsymbol{g}_t\|^2_{\text{diag}(\boldsymbol{\beta})^{-1}} - \frac{1}{2}D_\varphi(\boldsymbol{\beta},\boldsymbol{\eta}_t)$$

$$= -\frac{1}{2}\sum_{j=1}^{d}\left(\frac{g_{t,j}^2}{\beta_j} + \eta_{t,j}\varphi\left(\frac{\beta_j}{\eta_{t,j}}\right)\right).$$

The partial derivative of $\Psi_{t,\boldsymbol{x}}(\boldsymbol{\beta})$ with respect to $\beta_j$ is

$$\frac{\partial\Psi_{t,\boldsymbol{x}}(\boldsymbol{\beta})}{\partial\beta_j} = \frac{1}{2}\left(\frac{g_{t,j}^2}{\beta_j^2} - \varphi'\left(\frac{\beta_j}{\eta_{t,j}}\right)\right).$$

Note that $\varphi$ is a convex function, so $\varphi'$ is a non-decreasing function, and $\frac{\partial\Psi_{t,\boldsymbol{x}}(\boldsymbol{\beta})}{\partial\beta_j}$ is a non-increasing function. Recall that $\boldsymbol{\beta}^*$ be the solution of unconstrained problem $\max_{\boldsymbol{\beta}}(\min_{\boldsymbol{x}}\Psi_t(\boldsymbol{x},\boldsymbol{\beta}))$, hence, $\beta_j^*$ is a zero of function $\frac{\partial\Psi_{t,\boldsymbol{x}}(\boldsymbol{\beta})}{\partial\beta_j}$.

Moreover, if $\beta_j^* > B_{t,j}$, we have $\frac{\partial\Psi_{t,\boldsymbol{x}}(\boldsymbol{\beta})}{\partial\beta_j} \geq 0$. Thus, $\Psi_{t,\boldsymbol{x}}(\boldsymbol{\beta})$ with respect to $\beta_j$ is a non-decreasing function, and $\arg\max_{\beta_j}\Psi_{t,\boldsymbol{x}}(\boldsymbol{\beta}) = B_{t,j}$. For a similar reason, if $\beta_j^* < b_{t,j}$, then $\arg\max_{\beta_j}\Psi_{t,\boldsymbol{x}}(\boldsymbol{\beta}) = b_{t,j}$. In conclusion,

$$\arg\max_{\beta_j\in[b_{t,j},B_{t,j}]}\Psi_{t,\boldsymbol{x}}(\boldsymbol{\beta}) = \min\{\max\{\beta_j^*,b_{t,j}\},B_{t,j}\}, \text{ for } j = 1,\cdots,d.$$

$\square$

## F    PROOF OF LEMMA 5

In this section, we denote that $\Psi_{t,\boldsymbol{x}}(\beta) = \min_{\boldsymbol{x}\in\mathcal{X}}\Psi_t(\boldsymbol{x},\beta)$.

**Lemma 8.** $\beta_{t+1}$ *obtained from equation (9) satisfies* $\beta_{t+1} \geq \eta_t$.

*Proof.* Recall that $\varphi(1) = \varphi'(1) = 0$, so $\varphi(x) \geq 0$ for all $x$ and $D_\varphi(\beta, \eta_t) = \eta_t \varphi(\beta/\eta_t) \geq 0$. If $\beta < \eta_t$, then for all $\boldsymbol{x} \in \mathcal{X}$

$$\Psi_t(\boldsymbol{x}, \beta) = \boldsymbol{g}_t^\top (\boldsymbol{x} - \boldsymbol{x}_t) + \frac{\beta}{2} \|\boldsymbol{x} - \boldsymbol{x}_t\|_2^2 - \frac{1}{2} D_\varphi(\beta, \eta_t)$$
$$< \boldsymbol{g}_t^\top (\boldsymbol{x} - \boldsymbol{x}_t) + \frac{\eta_t}{2} \|\boldsymbol{x} - \boldsymbol{x}_t\|_2^2$$
$$= \Psi_t(\boldsymbol{x}, \eta_t).$$

Hence, $\min_{\boldsymbol{x} \in \mathcal{X}} \Psi_t(\boldsymbol{x}, \beta) < \min_{\boldsymbol{x} \in \mathcal{X}} \Psi_t(\boldsymbol{x}, \eta_t)$, i.e., $\Psi_{t,\boldsymbol{x}}(\beta) < \Psi_{t,\boldsymbol{x}}(\eta_t)$.
It means $\beta_{t+1} = \arg\max_{\beta \in \mathcal{B}} \Psi_{t,\boldsymbol{x}}(\beta) \geq \eta_t$. □

**Lemma 9.** $\beta_{t+1}$ *obtained from equation (11) satisfies* $\beta_{t+1} \geq \eta_t$.

*Proof.* Let $\boldsymbol{y} = \arg\min_{\boldsymbol{x}} \Psi(\boldsymbol{x}, \eta_t)$. If $\beta < \eta_t$, then

$$\Psi_t(\boldsymbol{y}, \beta) = \boldsymbol{g}_t^\top (\boldsymbol{y} - \boldsymbol{x}_t) + \frac{\beta}{2} \|\boldsymbol{y} - \boldsymbol{x}_t\|_2^2 - \frac{1}{2} D_\varphi(\beta, \eta_t)$$
$$< \boldsymbol{g}_t^\top (\boldsymbol{y} - \boldsymbol{x}_t) + \frac{\eta_t}{2} \|\boldsymbol{y} - \boldsymbol{x}_t\|_2^2$$
$$= \Psi_t(\boldsymbol{y}, \eta_t).$$

Hence $\beta_{t+1} = \arg\max_{\beta \in \mathcal{B}} \Psi_t(\boldsymbol{y}, \beta) \geq \eta_t$. □

## G  CONVERGENCE RATES IN ONLINE LEARNING SETTING

Recall the define of regret

$$R(T) = \sum_{t=0}^{T-1} (f_t(\boldsymbol{x}_t) - f_t(\boldsymbol{x}^*)), \tag{15}$$

where $\boldsymbol{x}^* = \arg\min_{\boldsymbol{x} \in \mathcal{X}} \sum_{t=0}^{T-1} f_t(\boldsymbol{x})$. We show our hyper-regularizer method with three update rules (Algorithm 1, 2 and 3) have $\mathcal{O}(\sqrt{T})$ regret bounds.

**Lemma 10** (Lemma 4 in Adagrad). *Consider an arbitrary real-valued sequence $\{a_i\}$ and its vector representation $a_{1:i} = (a_1, \cdots, a_i)^\top$. Then*

$$\sum_{t=1}^T \frac{a_t^2}{\|a_{1:t}\|_2} \leq 2\|a_{1:T}\|_2 \tag{16}$$

*holds.*

*Proof.* Let us use induction on $T$ to prove inequality (10). For $T = 1$, the inequality trivially holds. Assume the bound (16) holds true for $T - 1$, in which case

$$\sum_{t=1}^T \frac{a_t^2}{\|a_{1:t}\|_2} \leq 2\|a_{1:T-1}\|_2 + \frac{a_T^2}{\|a_{1:T}\|_2}.$$

We denote $b_T = \sum_{t=1}^T a_t^2$ and have

$$2\|a_{1:T-1}\|_2 + \frac{a_T^2}{\|a_{1:T}\|_2} = 2\sqrt{b_T - a_T^2} + \frac{a_T^2}{\sqrt{b_T}}$$
$$\leq 2\sqrt{b_T - a_T^2 + \frac{a_T^4}{4b_T}} + \frac{a_T^2}{\sqrt{b_T}}$$
$$= 2\sqrt{b_T}.$$

□

**Lemma 11.** *Suppose the sequence $\{\boldsymbol{x}_t\}$ and sequence $\{\boldsymbol{\beta}_t\}$ satisfy $\boldsymbol{x}_{t+1} = \boldsymbol{x}_t - \boldsymbol{g}_t/\boldsymbol{\beta}_{t+1}$. Then the regret satisfies*

$$2R(T) \leq \sum_{t=0}^{T-1} \|\boldsymbol{g}_t\|^2_{\mathrm{diag}(\boldsymbol{\beta}_{t+1})^{-1}} + \sum_{t=0}^{T-1} \|\boldsymbol{x}_t - \boldsymbol{x}^*\|^2_{\mathrm{diag}(\boldsymbol{\beta}_{t+1} - \boldsymbol{\beta}_t)} + \|\boldsymbol{x}_0 - \boldsymbol{x}^*\|^2_{\mathrm{diag}(\boldsymbol{\beta}_0)}$$

*Proof.* Note that

$$\boldsymbol{x}_{t+1} = \boldsymbol{x}_t - \mathrm{diag}(\boldsymbol{\beta}_{t+1})^{-1}\boldsymbol{g}_t,$$

and

$$\begin{aligned} &\|\boldsymbol{x}_{t+1} - \boldsymbol{x}^*\|^2_{\mathrm{diag}(\boldsymbol{\beta}_{t+1})} \\ &= \|\boldsymbol{x}_t - \boldsymbol{x}^* - \mathrm{diag}(\boldsymbol{\beta}_{t+1})^{-1}\boldsymbol{g}_t\|^2_{\mathrm{diag}(\boldsymbol{\beta}_{t+1})} \\ &= \|\boldsymbol{x}_t - \boldsymbol{x}^*\|^2_{\mathrm{diag}(\boldsymbol{\beta}_{t+1})} + \|\boldsymbol{g}_t\|^2_{\mathrm{diag}(\boldsymbol{\beta}_{t+1})^{-1}} - 2\boldsymbol{g}_t^\top(\boldsymbol{x}_t - \boldsymbol{x}^*), \end{aligned}$$

i.e.,

$$2\boldsymbol{g}_t^\top(\boldsymbol{x}_t - \boldsymbol{x}^*) = \|\boldsymbol{g}_t\|^2_{\mathrm{diag}(\boldsymbol{\beta}_{t+1})^{-1}} + \left( \|\boldsymbol{x}_t - \boldsymbol{x}^*\|^2_{\mathrm{diag}(\boldsymbol{\beta}_{t+1})} - \|\boldsymbol{x}_{t+1} - \boldsymbol{x}^*\|^2_{\mathrm{diag}(\boldsymbol{\beta}_{t+1})} \right). \quad (17)$$

Hence

$$\begin{aligned} 2R(T) &= 2\sum_{t=0}^{T-1}(f_t(\boldsymbol{x}_t) - f_t(\boldsymbol{x}_*)) \\ &\leq 2\sum_{t=0}^{T-1}\boldsymbol{g}_t^\top(\boldsymbol{x}_t - \boldsymbol{x}^*) \\ &= \sum_{t=0}^{T-1}\|\boldsymbol{g}_t\|^2_{\mathrm{diag}(\boldsymbol{\beta}_{t+1})^{-1}} + \sum_{t=0}^{T-1}\left( \|\boldsymbol{x}_t - \boldsymbol{x}^*\|^2_{\mathrm{diag}(\boldsymbol{\beta}_{t+1})} - \|\boldsymbol{x}_{t+1} - \boldsymbol{x}^*\|^2_{\mathrm{diag}(\boldsymbol{\beta}_{t+1})} \right) \\ &\leq \sum_{t=0}^{T-1}\|\boldsymbol{g}_t\|^2_{\mathrm{diag}(\boldsymbol{\beta}_{t+1})^{-1}} + \sum_{t=0}^{T-1}\|\boldsymbol{x}_t - \boldsymbol{x}^*\|^2_{\mathrm{diag}(\boldsymbol{\beta}_{t+1} - \boldsymbol{\beta}_t)} + \|\boldsymbol{x}_0 - \boldsymbol{x}^*\|^2_{\mathrm{diag}(\boldsymbol{\beta}_0)}. \end{aligned}$$

$\square$

**Lemma 12.** *Suppose an increasing function $\psi$ satisfies $\psi(1) = 0$ and $\psi(x) \leq l(x-1)$. Consider a real valued sequence $\{g_t\}_{t=0:T-1}$ and a positive sequence $\{\beta_t\}_{t=0:T}$ which satisfies $|g_t| \leq G$, $\beta_{t+1}^2\psi\left(\frac{\beta_{t+1}}{\beta_t}\right) = g_t^2$, $t = 0,\cdots,T-1$, $\beta_0 \geq 0$. We can bound $\beta_T$ as*

$$\beta_t \geq c\sqrt{\beta_0^2 + \frac{2}{l}\sum_{i=0}^{t-1}g_i^2}, \ t = 1,\cdots,T \quad (18)$$

*where $c = \sqrt{\frac{\beta_0^2}{\beta_0^2 + 2G^2/l}}$. Moreover, we have*

$$\sum_{t=0}^{T-1}\frac{g_t^2}{\beta_{t+1}} \leq \frac{\sqrt{2l\beta_0^2 + 4G^2}}{\beta_0}\sqrt{\sum_{t=0}^{T-1}g_t^2}. \quad (19)$$

**Remark.** *We point out that*

- *$\beta_{t+1} \geq \beta_t$ (If $\beta_{t+1} < \beta_t$, then $\beta_{t+1}^2\psi(\beta_{t+1}/\beta_t) < 0 \leq g_t^2$),*

- *$\beta_{t+1}$ is unique with respect to $\beta_t$ due to the fact that the function $\hat{\psi}(\beta) = \beta^2\psi(\beta/\beta_t)$ is strictly increasing.*

*Proof.* Assume that $\beta_t \geq c\sqrt{\beta_0^2 + \frac{2}{l}\sum_{i=0}^{t-1} g_i^2}$, where $c > 0$ is a variable coefficient.

Let us find out a specific $c$ such that $\beta_{t+1} \geq c\sqrt{\beta_0^2 + \frac{2}{l}\sum_{i=0}^{t} g_i^2}$.

Note that

$$g_t^2 = \beta_{t+1}^2 \psi\left(\frac{\beta_{t+1}}{\beta_t}\right) \leq l\beta_{t+1}^2\left(\frac{\beta_{t+1}}{\beta_t} - 1\right). \tag{20}$$

Define a cubic polynomial

$$h(\beta) = \frac{l}{\beta_t}\beta^3 - l\beta^2 - g_t^2,$$

and $h$ is an increasing function when $\beta \geq \beta_t$.

If $h\left(c\sqrt{\beta_0^2 + \frac{2}{l}\sum_{i=0}^{t} g_i^2}\right) \leq 0$, according to $h(\beta_{t+1}) \geq 0$, then $\beta_{t+1} \geq c\sqrt{\beta_0^2 + \frac{2}{l}\sum_{i=0}^{t} g_i^2}$.

Denote $b = \beta_0^2 + \frac{2}{l}\sum_{i=0}^{t-1} g_i^2$. So we just need to choose $c$ such that

$$h\left(c\sqrt{\beta_0^2 + \frac{2}{l}\sum_{i=0}^{t} g_i^2}\right) \leq lc^2(b + 2g_t^2/l)\left(\frac{\sqrt{b + 2g_t^2/l}}{\sqrt{b}} - 1\right) - g_t^2 \leq 0,$$

where the first inequality holds for the assumption $\beta_t \geq c\sqrt{\beta_0^2 + \frac{2}{l}\sum_{i=0}^{t-1} g_i^2}$, or

$$\frac{c^2}{\sqrt{b}}(b + 2g_t^2/l)\frac{2g_t^2/l}{\sqrt{b + 2g_t^2/l} + \sqrt{b}} \leq g_t^2/l,$$

or

$$\frac{2c^2}{\sqrt{b}}(b + 2g_t^2/l) \leq \sqrt{b + 2g_t^2/l} + \sqrt{b}.$$

Thus, $c$ just need to satisfy

$$c^2 \leq \frac{b}{b + 2g_t^2/l}.$$

According to $b \geq \beta_0^2$, $g_t^2 \leq G^2$, hence

$$\frac{b}{b + 2g_t^2/l} \geq \frac{\beta_0^2}{\beta_0^2 + 2G^2/l}.$$

So if we choose $c = \sqrt{\frac{\beta_0^2}{\beta_0^2 + 2G^2/l}}$, then $\beta_1 > \beta_0 > c\beta_0$, hence

$$\beta_t \geq c\sqrt{\beta_0^2 + \frac{2}{l}\sum_{i=0}^{t-1} g_i^2}, t = 1, \cdots, T.$$

Moreover, following from Lemma 10, we have

$$\sum_{t=0}^{T-1} \frac{g_t^2}{\beta_{t+1}} \leq \sum_{t=0}^{T-1} \frac{g_t^2}{c\sqrt{2/l}\sqrt{\sum_{i=0}^{t} g_i^2}} \leq \frac{\sqrt{2l}}{c}\sqrt{\sum_{t=0}^{T-1} g_t^2}.$$

$\square$

**Lemma 13.** *Suppose an increasing function $\psi$ satisfies $\psi(1) = 0$ and $\psi(x) \leq l(x - 1)$. Consider a real valued sequence $\{g_t\}_{t=0:T-1}$ and a positive sequence $\{\beta_t\}_{t=0:T}$ which satisfies $|g_t| \leq G$, $\beta_t^2 \psi\left(\frac{\beta_{t+1}}{\beta_t}\right) = g_t^2$, $t = 0, \cdots, T-1$, $\beta_0 \geq 0$. We can bound $\beta_T$ as*

$$\beta_t \geq \sqrt{\beta_0^2 + \frac{2}{l}\sum_{i=0}^{t-1} g_i^2}, t = 1, \cdots, T. \tag{21}$$

*Moreover, we have*

$$\sum_{t=0}^{T-1} \frac{g_t^2}{\beta_t} \leq \max\left\{\sqrt{2l}, \frac{2G}{\beta_0}\right\}\sqrt{\sum_{t=0}^{T-1} g_t^2}. \tag{22}$$

*Proof.* Same as inequality (20), we have

$$l\beta_t^2\left(\frac{\beta_{t+1}}{\beta_t} - 1\right) \geq g_t^2,$$

hence

$$\beta_{t+1}^2 = \left(\beta_t + \frac{g_t^2}{l\beta_t}\right)^2 \geq \beta_t^2 + \frac{2}{l}g_t^2 \geq \beta_0^2 + \frac{2}{l}\sum_{i=0}^{t} g_t^2 \geq \min\left\{1, \frac{l\beta_0^2}{2G^2}\right\}\frac{2}{l}\sum_{i=0}^{t+1} g_i^2,.$$

Furthermore, following from Lemma 10, we have

$$\sum_{t=0}^{T-1}\frac{g_t^2}{\beta_t} \leq \sqrt{\frac{l/2}{\min\{1, l\beta_0^2/(2G^2)\}}}\sum_{t=0}^{T-1}\frac{g_t^2}{\sqrt{\sum_{i=0}^{t} g_i^2}} \leq \max\left\{\sqrt{2l}, \frac{2G}{\beta_0}\right\}\sqrt{\sum_{t=0}^{T-1} g_t^2}.$$

$\square$

**Theorem 14.** *Suppose that $\varphi \in C_l^{1,1}([1, +\infty))$, and $\varphi$ is $\alpha$-strongly convex. Assume that $\|\boldsymbol{g}_t\|_\infty \leq G$, and $\|\boldsymbol{x}_t - \boldsymbol{x}^*\|_\infty \leq D_\infty$. Then the sequence $\{\boldsymbol{x}_t\}$ obtained from Algorithm 1 satisfies*

$$2R(T) \leq \frac{(\alpha + D_\infty^2)\sqrt{2l\beta_0^2 + 4G^2}}{\alpha\beta_0}\sum_{j=1}^{d}\|g_{0:T-1,j}\|_2 + \beta_0\|\boldsymbol{x}_0 - \boldsymbol{x}^*\|_2^2,$$

*Proof.* Following from Lemma 11,

$$2R(T) \leq \sum_{t=0}^{T-1}\|\boldsymbol{g}_t\|_{\mathrm{diag}(\boldsymbol{\beta}_{t+1})^{-1}}^2 + \sum_{t=0}^{T-1}\|\boldsymbol{x}_t - \boldsymbol{x}^*\|_{\mathrm{diag}(\boldsymbol{\beta}_{t+1}-\boldsymbol{\beta}_t)}^2 + \|\boldsymbol{x}_0 - \boldsymbol{x}^*\|_{\mathrm{diag}(\boldsymbol{\beta}_0)}^2$$

$$\leq \sum_{t=0}^{T-1}\|\boldsymbol{g}_t\|_{\mathrm{diag}(\boldsymbol{\beta}_{t+1})^{-1}}^2 + \sum_{t=0}^{T-1}\|\boldsymbol{x}_t - \boldsymbol{x}^*\|_\infty^2\|\boldsymbol{\beta}_{t+1} - \boldsymbol{\beta}_t\|_1 + \|\boldsymbol{x}_0 - \boldsymbol{x}^*\|_{\mathrm{diag}(\boldsymbol{\beta}_0)}^2$$

$$\leq \sum_{t=0}^{T-1}\sum_{j=1}^{d}\frac{g_{t,j}^2}{\beta_{t+1,j}} + \max_{0\leq t<T}\|\boldsymbol{x}_t - \boldsymbol{x}^*\|_\infty^2\sum_{t=0}^{T-1}\sum_{j=1}^{d}(\beta_{t+1,j} - \beta_{t,j}) + \|\boldsymbol{x}_0 - \boldsymbol{x}^*\|_{\mathrm{diag}(\boldsymbol{\beta}_0)}^2.$$

Recall $\varphi$ is a $\alpha$-strongly convex function, so,

$$g_{t,j}^2 = \beta_{t+1,j}^2\varphi'\left(\frac{\beta_{t+1,j}}{\beta_{t,j}}\right) \geq \alpha\beta_{t+1,j}\beta_{t,j}\left(\frac{\beta_{t+1,j}}{\beta_{t,j}} - 1\right),$$

and

$$\sum_{t=0}^{T-1}(\beta_{t+1,j} - \beta_{t,j}) \leq \frac{1}{\alpha}\sum_{t=0}^{T-1}\frac{g_{t,j}^2}{\beta_{t+1,j}}. \tag{23}$$

The function $\psi = \varphi'$ satisfies $\psi(1) = 0$ and $\psi(x) \leq l(x - 1)$ according to the smoothness of $\varphi$. Following from Lemma 12, we have

$$\sum_{t=0}^{T-1}\frac{g_{t,j}^2}{\beta_{t+1,j}} \leq \frac{\sqrt{2l\beta_{0,j}^2 + 4G^2}}{\beta_{0,j}}\sqrt{\sum_{i=0}^{T-1} g_{t,j}^2} = \frac{\sqrt{2l\beta_{0,j}^2 + 4G^2}}{\beta_{0,j}}\|g_{0:T-1,j}\|_2. \tag{24}$$

Combining inequality (23) and (24), we have

$$2R(T) \leq \left(1 + \frac{\max_{0\leq t<T}\|\boldsymbol{x}_t - \boldsymbol{x}^*\|_\infty^2}{\alpha}\right)\sum_{j=1}^{d}\sum_{t=0}^{T-1}\frac{g_{t,j}^2}{\beta_{t+1,j}} + \|\boldsymbol{x}_0 - \boldsymbol{x}^*\|_{\mathrm{diag}(\boldsymbol{\beta}_0)}^2$$

$$\leq (1 + \frac{D_\infty^2}{\alpha})\sum_{j=1}^{d}\frac{\sqrt{2l\beta_{0,j}^2 + 4G^2}}{\beta_{0,j}}\|g_{0:T-1,j}\|_2 + \|\boldsymbol{x}_0 - \boldsymbol{x}^*\|_{\mathrm{diag}(\boldsymbol{\beta}_0)}^2$$

$$= \frac{(\alpha + D_\infty^2)\sqrt{2l\beta_0^2 + 4G^2}}{\alpha\beta_0}\sum_{j=1}^{d}\|g_{0:T-1,j}\|_2 + \beta_0\|\boldsymbol{x}_0 - \boldsymbol{x}^*\|_2^2.$$

$\square$

**Theorem 15.** *Suppose that $\varphi \in C_l^{1,1}\left([1, +\infty)\right)$, and $\varphi$ is $\alpha$-strongly convex. Assume that $\|\boldsymbol{g}_t\|_\infty \leq G$, and $\|\boldsymbol{x}_t - \boldsymbol{x}^*\|_\infty \leq D_\infty$. Then the sequence $\{\boldsymbol{x}_t\}$ obtained from Algorithm 2 (or 3) satisfies*

$$2R(T) \leq \left(1 + \frac{D_\infty^2}{\alpha}\right) \max\left\{\sqrt{2l}, \frac{2G}{\beta_0}\right\} \sum_{j=1}^{d} \|g_{0:T-1,j}\|_2 + \beta_0 \|\boldsymbol{x}_0 - \boldsymbol{x}^*\|_2^2.$$

*Proof.* Similar to the proof of Theorem 14, for Algorithm 2, we have

$$2R(T) \leq \sum_{t=0}^{T-1} \sum_{j=1}^{d} \frac{g_{t,j}^2}{\beta_{t+1,j}} + \max_{0 \leq t < T} \|\boldsymbol{x}_t - \boldsymbol{x}^*\|_\infty^2 \sum_{t=0}^{T-1} \sum_{j=1}^{d} (\beta_{t+1,j} - \beta_{t,j}) + \|\boldsymbol{x}_0 - \boldsymbol{x}^*\|_{\mathrm{diag}(\boldsymbol{\beta}_0)}^2$$

$$\leq \sum_{t=0}^{T-1} \sum_{j=1}^{d} \frac{g_{t,j}^2}{\beta_{t,j}} + \max_{0 \leq t < T} \|\boldsymbol{x}_t - \boldsymbol{x}^*\|_\infty^2 \sum_{t=0}^{T-1} \sum_{j=1}^{d} (\beta_{t+1,j} - \beta_{t,j}) + \|\boldsymbol{x}_0 - \boldsymbol{x}^*\|_{\mathrm{diag}(\boldsymbol{\beta}_0)}^2.$$

With same reason, for Algorithm 3, we have

$$2R(T) \leq \sum_{t=0}^{T-1} \|\boldsymbol{g}_t\|_{\mathrm{diag}(\boldsymbol{\beta}_t)^{-1}}^2 + \sum_{t=0}^{T-1} \left(\|\boldsymbol{x}_t - \boldsymbol{x}^*\|_{\mathrm{diag}(\boldsymbol{\beta}_t)}^2 - \|\boldsymbol{x}_{t+1} - \boldsymbol{x}^*\|_{\mathrm{diag}(\boldsymbol{\beta}_t)}^2\right)$$

$$\leq \sum_{t=0}^{T-1} \|\boldsymbol{g}_t\|_{\mathrm{diag}(\boldsymbol{\beta}_t)^{-1}}^2 + \sum_{t=1}^{T-1} \|\boldsymbol{x}_t - \boldsymbol{x}^*\|_{\mathrm{diag}(\boldsymbol{\beta}_t - \boldsymbol{\beta}_{t-1})}^2 + \|\boldsymbol{x}_0 - \boldsymbol{x}^*\|_{\mathrm{diag}(\boldsymbol{\beta}_0)}^2$$

$$\leq \sum_{t=0}^{T-1} \sum_{j=1}^{d} \frac{g_{t,j}^2}{\beta_{t,j}} + \max_{0 \leq t < T} \|\boldsymbol{x}_t - \boldsymbol{x}^*\|_\infty^2 \sum_{t=1}^{T-1} \sum_{j=1}^{d} (\beta_{t,j} - \beta_{t-1,j}) + \|\boldsymbol{x}_0 - \boldsymbol{x}^*\|_{\mathrm{diag}(\boldsymbol{\beta}_0)}^2,$$

Note that for both algorithms

$$g_{t,j}^2 = \beta_{t,j}^2 \varphi'\left(\frac{\beta_{t+1,j}}{\beta_{t,j}}\right) \geq \alpha \beta_{t,j}^2 \left(\frac{\beta_{t+1,j}}{\beta_{t,j}} - 1\right),$$

holds, so

$$\sum_{t=1}^{T-1} (\beta_{t,j} - \beta_{t-1,j}) \leq \sum_{t=0}^{T-1} (\beta_{t+1,j} - \beta_{t,j}) \leq \frac{1}{\alpha} \sum_{t=0}^{T-1} \frac{g_{t,j}^2}{\beta_{t,j}},$$

Thus, following from Lemma 13, for both Algorithm 2 and 3, we have

$$2R(T) \leq (1 + \frac{D_\infty^2}{\alpha}) \sum_{j=1}^{d} \sum_{t=0}^{T-1} \frac{g_{t,j}^2}{\beta_{t,j}} + \beta_0 \|\boldsymbol{x}_0 - \boldsymbol{x}^*\|_2^2$$

$$\leq (1 + \frac{D_\infty^2}{\alpha}) \sum_{j=1}^{d} \max\left\{\sqrt{2l}, \frac{2G}{\beta_0}\right\} \|g_{0:T-1,j}\|_2 + \beta_0 \|\boldsymbol{x}_0 - \boldsymbol{x}^*\|_2^2.$$

$\square$

# H LOGARITHMIC BOUNDS

In this section, we will use a different class of 'distance' function for problem 4, and establish logarithmic regret bounds under assumption $f_t$ is strongly convex. Our analysis and proof follow from Hazan et al. (2007); Mukkamala & Hein (2017).

First, we define $\boldsymbol{\mu}$-strongly convexity.

**Definition 16** (Definition 2.1 in (Mukkamala & Hein, 2017)). *Let $\mathcal{X} \subseteq \mathbb{R}^d$ be a convex set. We say that a function $f : \mathcal{X} \to \mathbb{R}$ is $\boldsymbol{\mu}$-strongly convex, if there exists $\boldsymbol{\mu} \in \mathbb{R}^d$ with $\mu_j > 0$ for $j = 1, \cdots, d$ such that for all $\boldsymbol{x}, \boldsymbol{y} \in \mathcal{X}$,*

$$f(\boldsymbol{y}) \geq f(\boldsymbol{x}) + \langle \nabla f(\boldsymbol{x}), \boldsymbol{y} - \boldsymbol{x} \rangle + \frac{1}{2} \|\boldsymbol{y} - \boldsymbol{x}\|_{\mathrm{diag}(\boldsymbol{\mu})}^2.$$

*Let $\xi = \min_{j=1:d} \mu_j$, then this function is $\xi$-strongly convex (in the usual sense), that is*

$$f(\boldsymbol{y}) \geq f(\boldsymbol{x}) + \langle \nabla f(\boldsymbol{x}), \boldsymbol{y} - \boldsymbol{x} \rangle + \frac{\xi}{2} \|\boldsymbol{y} - \boldsymbol{x}\|_2^2.$$

The modification SC-Hyper-Regularization of Hyper-Regularization which we propose in the following uses a family of distance function $D : \mathbb{R}_{++}^d \times \mathbb{R}_{++}^d \to \mathbb{R}$ formulated as

$$D(\boldsymbol{u}, \boldsymbol{v}) = \sum_{j=1}^d \varphi(u_j/v_j), \tag{25}$$

where $\varphi$ is convex function with $\varphi(1) = \varphi'(1) = 0$ like we used in $\varphi$-divergence.

**Remark.** *Same as $\varphi$-divergence, $D(\boldsymbol{u}, \boldsymbol{v}) \geq 0$ for any $\boldsymbol{u}, \boldsymbol{v} \in \mathbb{R}_{++}^d$.*

Different from Algorithm 1 and 2, we add a hyper-parameter $\lambda > 0$ like AdaGrad to SC-Hyper-Gradient. Rewrite problem (5) as

$$\max_{\boldsymbol{\beta} \in \mathcal{B}_t} \min_{\boldsymbol{x} \in \mathcal{X}} \Psi_t(\boldsymbol{x}, \boldsymbol{\beta}) \triangleq \boldsymbol{g}_t^\top (\boldsymbol{x} - \boldsymbol{x}_t) + \frac{1}{2}\|\boldsymbol{x} - \boldsymbol{x}_t\|_{\mathrm{diag}(\boldsymbol{\beta})}^2 - \frac{\lambda}{2}\sum_{j=1}^d \varphi(\beta_j/\beta_{t,j}), \tag{26}$$

and corresponding two algorithms as

---

**Algorithm 4** GD with SC-Hyper-regularization

---

**Input:** $\boldsymbol{\beta}_0 > 0$, $\boldsymbol{x}_0$
1: **for** $t = 1$ to $T$ **do**
2:     Suffer loss $f_t(\boldsymbol{x}_t)$;
3:     Receive $\boldsymbol{g}_t \in \partial f_t(\boldsymbol{x}_t)$ of $f_t$ at $\boldsymbol{x}_t$;
4:     Update $\beta_{t+1,j}$ as the solution of the equation $\lambda(\beta^2/\beta_{t,j})\varphi'(\beta/\beta_{t,j}) = g_{t,j}^2, j = 1, \cdots, d$;
5:     Update $\boldsymbol{x}_{t+1} = \boldsymbol{x}_t - \boldsymbol{g}_t/\boldsymbol{\beta}_{t+1}$;
6: **end for**

---

**Algorithm 5** GD with SC-Hyper-regularization using alternating update rule

---

**Input:** $\boldsymbol{\beta}_0 > 0$, $\boldsymbol{x}_0$
1: **for** $t = 1$ to $T$ **do**
2:     Suffer loss $f_t(\boldsymbol{x}_t)$;
3:     Receive $\boldsymbol{g}_t \in \partial f_t(\boldsymbol{x}_t)$ of $f_t$ at $\boldsymbol{x}_t$;
4:     Update $\beta_{t+1,j} = \beta_{t,j}(\varphi')^{-1}(g_{t,j}^2/(\lambda\beta_{t,j})), j = 1, \ldots, d$;
5:     Update $\boldsymbol{x}_{t+1} = \boldsymbol{x}_t - \boldsymbol{g}_t/\boldsymbol{\beta}_{t+1}$;
6: **end for**

---

**Remark.** *Same as Lemma 5, the monotonicity of Algorithm 4 and 5 also holds.*

**Theorem 17.** *Suppose that $f_t$ is $\boldsymbol{\mu}$-strongly convex for all $t$, $\varphi \in C_l^{1,1}([1, +\infty))$, and $\varphi$ is $\alpha$-strongly convex. Assume that $\|\boldsymbol{g}_t\|_\infty \leq G$, and $\lambda \geq G^2/(\alpha \min_{j=1:d} \mu_j)$. Then the sequence $\{\boldsymbol{x}_t\}$ obtained from Algorithm 4 satisfies*

$$2R(T) \leq l\left(1 + \frac{G^2}{\lambda l \beta_0}\right)^2 \sum_{j=1}^d \ln\left(1 + \frac{\|g_{0:T-1,j}\|_2^2}{\lambda l \beta_0}\right) + \beta_0\|\boldsymbol{x}_0 - \boldsymbol{x}^*\|_2^2,$$

*and the sequence $\{\boldsymbol{x}_t\}$ obtained from Algorithm 5 satisfies*

$$2R(T) \leq l\sum_{j=1}^d \ln\left(1 + \frac{\|g_{0:T-1,j}\|_2^2}{\lambda l \beta_0}\right) + \beta_0\|\boldsymbol{x}_0 - \boldsymbol{x}^*\|_2^2,$$

**Remark.** *Under assumption in Theorem 17, we have $\|g_{0:T-1,j}\|_2^2 \leq G^2 T$, so $R(T) = \mathcal{O}(\ln(T))$.*

To prove Theorem 17, we first prove following lemma.

**Lemma 18.** *For an arbitrary real-valued sequence $\{a_i\}$ and a positive real number $b$,*

$$\sum_{t=1}^T \frac{a_t^2}{b + \sum_{i=1}^t a_i^2} \leq \ln\left(1 + \frac{\sum_{t=1}^T a_t^2}{b}\right). \tag{27}$$

*Proof.* Let $b_0 = b, b_t = b + \sum_{i=1}^{t} a_i^2, t \geq 1$, then

$$
\sum_{t=1}^{T} \frac{a_t^2}{b + \sum_{i=1}^{t} a_i^2} = \sum_{t=1}^{T} \frac{b_t - b_{t-1}}{b_t} = \sum_{t=1}^{T} \int_{b_{t-1}}^{b_t} \frac{1}{b_t} dx
$$

$$
\leq \sum_{t=1}^{T} \int_{b_{t-1}}^{b_t} \frac{1}{x} dx = \int_{b}^{b_T} \frac{1}{x} dx = \ln\left(1 + \frac{\sum_{t=1}^{T} a_t^2}{b}\right).
$$

$\square$

Like Lemma 12 and 13, similar lemma holds for Algorithm 4 and 5.

**Lemma 19.** *Suppose an increasing function $\psi$ satisfies $\psi(1) = 0$ and $\psi(x) \leq l(x - 1)$. Consider a real valued sequence $\{g_t\}_{t=0:T-1}$ and a positive sequence $\{\beta_t\}_{t=0:T}$ which satisfies $|g_t| \leq G$, $\beta_0 > 0$.*
*If $(\beta_{t+1}^2/\beta_t)\psi(\beta_{t+1}/\beta_t) = g_t^2$, $t = 0, \cdots, T-1$, then we have*

$$
\beta_t \geq \left(\frac{\beta_0}{\beta_0 + G^2/l}\right)^2 \left(\beta_0 + \frac{1}{l}\sum_{i=0}^{t-1} g_i^2\right), \ t = 1, \cdots, T \tag{28}
$$

*and*

$$
\sum_{t=0}^{T-1} \frac{g_t^2}{\beta_{t+1}} \leq l\left(\frac{\beta_0 + G^2/l}{\beta_0}\right)^2 \ln\left(1 + \frac{\sum_{t=0}^{T-1} g_t^2}{l\beta_0}\right). \tag{29}
$$

*Meanwhile, if $\beta_t \psi(\beta_{t+1}/\beta_t) = g_t^2$, $t = 0, \cdots, T-1$, then we have*

$$
\beta_t \geq \beta_0 + \frac{1}{l}\sum_{i=0}^{t-1} g_i^2, \ t = 1, \cdots, T \tag{30}
$$

*and*

$$
\sum_{t=0}^{T-1} \frac{g_t^2}{\beta_{t+1}} \leq l \ln\left(1 + \frac{\sum_{t=0}^{T-1} g_t^2}{l\beta_0}\right). \tag{31}
$$

*Proof.* Using same methods in proof of Lemma 12 and 13, the conclusion can be deduced from Lemma 18 easily. $\square$

***proof of Theorem 17.*** Like Lemma 11, in strongly convex case, we have

$$
2R(T) = 2\sum_{t=0}^{T-1} f_t(\boldsymbol{x}_t) - f_t(\boldsymbol{x}^*)
$$

$$
\leq 2\sum_{t=0}^{T-1} \langle \boldsymbol{g}_t, \boldsymbol{x}_t - \boldsymbol{x}^* \rangle - \sum_{t=0}^{T-1} \|\boldsymbol{x}_t - \boldsymbol{x}^*\|_{\text{diag}(\boldsymbol{\mu})}^2
$$

$$
= \sum_{t=0}^{T-1} \|\boldsymbol{g}_t\|_{\text{diag}(\boldsymbol{\beta}_{t+1})^{-1}}^2 + \sum_{t=0}^{T-1} \left(\|\boldsymbol{x}_t - \boldsymbol{x}^*\|_{\text{diag}(\boldsymbol{\beta}_{t+1})}^2 - \|\boldsymbol{x}_{t+1} - \boldsymbol{x}^*\|_{\text{diag}(\boldsymbol{\beta}_{t+1})}^2\right) - \sum_{t=0}^{T-1} \|\boldsymbol{x}_t - \boldsymbol{x}^*\|_{\text{diag}(\boldsymbol{\mu})}^2
$$

$$
\leq \sum_{t=0}^{T-1} \|\boldsymbol{g}_t\|_{\text{diag}(\boldsymbol{\beta}_{t+1})^{-1}}^2 + \sum_{t=0}^{T-1} \|\boldsymbol{x}_t - \boldsymbol{x}^*\|_{\text{diag}(\boldsymbol{\beta}_{t+1} - \boldsymbol{\beta}_t - \boldsymbol{\mu})}^2 + \|\boldsymbol{x}_0 - \boldsymbol{x}^*\|_{\text{diag}(\boldsymbol{\beta}_0)}^2.
$$

Note that in Algorithm 4,

$$
\beta_{t+1,j} - \beta_{t,j} = \beta_{t,j}(\beta_{t+1,j}/\beta_{t,j} - 1)
$$

$$
\leq \frac{\beta_{t,j}}{\alpha}\varphi'\left(\frac{\beta_{t+1,j}}{\beta_{t,j}}\right) = \frac{\beta_{t,j}^2}{\beta_{t+1,j}^2}\frac{g_{t,j}^2}{\lambda\alpha} \leq \frac{G^2}{\lambda\alpha}
$$

holds, and in Algorithm 5, same conclusion holds:

$$\beta_{t+1,j} - \beta_{t,j} = \beta_{t,j}(\beta_{t+1,j}/\beta_{t,j} - 1)$$

$$\leq \frac{\beta_{t,j}}{\alpha}\varphi'\left(\frac{\beta_{t+1,j}}{\beta_{t,j}}\right) = \frac{g_{t,j}^2}{\lambda\alpha} \leq \frac{G^2}{\lambda\alpha}.$$

Hence, if $\lambda \geq \max_{j=1:d}\frac{G^2}{\alpha\mu_j}$, then $\boldsymbol{\beta}_{t+1} - \boldsymbol{\beta}_t \leq \boldsymbol{\mu}$, and

$$\sum_{t=0}^{T-1}\|\boldsymbol{x}_t - \boldsymbol{x}^*\|_{\mathrm{diag}(\boldsymbol{\beta}_{t+1}-\boldsymbol{\beta}_t-\boldsymbol{\mu})}^2 \leq 0.$$

On the other hand,

$$\sum_{t=0}^{T-1}\|\boldsymbol{g}_t\|_{\mathrm{diag}(\boldsymbol{\beta}_{t+1})^{-1}}^2 = \sum_{j=1}^{d}\sum_{t=0}^{T-1}\frac{g_{t,j}^2}{\beta_{t+1,j}},$$

following from Lemma 19, we have

$$\sum_{t=0}^{T-1}\|\boldsymbol{g}_t\|_{\mathrm{diag}(\boldsymbol{\beta}_{t+1})^{-1}}^2 \leq l\left(1 + \frac{G^2}{\lambda l\beta_0}\right)^2\sum_{j=1}^{d}\ln\left(1 + \frac{\|g_{0:T-1,j}\|_2^2}{\lambda l\beta_0}\right) \quad \text{in Algorithm 4,}$$

$$\sum_{t=0}^{T-1}\|\boldsymbol{g}_t\|_{\mathrm{diag}(\boldsymbol{\beta}_{t+1})^{-1}}^2 \leq l\sum_{j=1}^{d}\ln\left(1 + \frac{\|g_{0:T-1,j}\|_2^2}{\lambda l\beta_0}\right) \quad \text{in Algorithm 5.}$$

$\square$

# I    CONVERGENCE RATES IN FULL BATCH SETTING

In this section, we will discuss the convergence of our methods in full batch settings.

We first review a classical result on the convergence rate for gradient descent with fixed learning rate.

**Theorem 20.** *Suppose that $F \in C_L^{1,1}(\mathbb{R}^d)$ and $F^* = \inf_{\boldsymbol{x}}F(\boldsymbol{x}) > -\infty$. Consider gradient descent with constant step size, $\boldsymbol{x}_{t+1} = \boldsymbol{x}_t - \frac{\nabla F(\boldsymbol{x}_t)}{b}$. If $b > \frac{L}{2}$, then*

$$\min_{0\leq t\leq T-1}\|\nabla F(\boldsymbol{x}_t)\|_2^2 \leq \varepsilon$$

*after at most a number of steps*

$$T = \frac{2b^2(F(\boldsymbol{x}_0) - F^*)}{\varepsilon(2b - L)} = \mathcal{O}\left(\frac{1}{\varepsilon}\right)$$

*Proof.* Following from the fact that $F$ is $L$-smooth, we have

$$F(\boldsymbol{x}_{t+1}) \leq F(\boldsymbol{x}_t) + \nabla F(\boldsymbol{x}_t)^\top(\boldsymbol{x}_{t+1} - \boldsymbol{x}_t) + \frac{L}{2}\|\boldsymbol{x}_{t+1} - \boldsymbol{x}_t\|_2^2$$

$$= F(\boldsymbol{x}_t) - \frac{1}{b}\|\nabla F(\boldsymbol{x}_t)\|_2^2 + \frac{L}{2b^2}\|\nabla F(\boldsymbol{x}_t)\|_2^2$$

$$= F(\boldsymbol{x}_t) - \frac{1}{b}\left(1 - \frac{L}{2b}\right)\|\nabla F(\boldsymbol{x}_t)\|_2^2. \tag{32}$$

When $b > \frac{L}{2}$, $1 - \frac{L}{2b} > 0$. So

$$\sum_{t=0}^{T-1}\|\nabla F(\boldsymbol{x}_t)\|_2^2 \leq \frac{2b^2}{2b-L}(F(\boldsymbol{x}_0) - F(\boldsymbol{x}_T)) \leq \frac{2b^2}{2b-L}(F(\boldsymbol{x}_0) - F^*),$$

and

$$\min_{0\leq t\leq T-1}\|\nabla F(\boldsymbol{x}_t)\|_2^2 \leq \frac{1}{T}\sum_{t=0}^{T-1}\|\nabla F(\boldsymbol{x}_t)\|_2^2 \leq \frac{2b^2}{T(2b-L)}(F(\boldsymbol{x}_0) - F^*) \leq \varepsilon.$$

$\square$

**Remark.** *If we choose* $b \leq \frac{L}{2}$*, then convergence of gradient descent with constant learning rate is not guaranteed at all.*

Like Algorithm 3, another update rule is worth considering in full batch setting:

$$\begin{cases} \boldsymbol{x}_{t+1} = \boldsymbol{x}_t - \frac{1}{\beta_t}\boldsymbol{g}_t, \\ \beta_{t+1} = \beta_t(\varphi')^{-1}(\|\boldsymbol{g}_t\|_2^2/\beta_t^2) \end{cases} \tag{33}$$

Next we will show that both update rules (13) and (14) are robust to the choice of initial learning rate. Our proof is followed from the proof of Theorem 2.3 in WNGrad (Wu et al., 2018). Note that in update rule (13), $\beta_{t+1}$ satisfies

$$\beta_{t+1}^2 \varphi'(\beta_{t+1}/\beta_t) = \|\boldsymbol{g}_t\|_2^2,$$

while in update rule (14) and (33), $\beta_{t+1}$ satisfies

$$\beta_t^2 \varphi'(\beta_{t+1}/\beta_t) = \|\boldsymbol{g}_t\|_2^2.$$

**Theorem 21** (Convergence rate of update rule (13)). *Suppose that* $\varphi \in C_l^{1,1}([1, +\infty))$*,* $\varphi$ *is* $\alpha$*-strongly convex, and* $F \in C_L^{1,1}(\mathbb{R}^d)$*,* $F^* = \inf_{\boldsymbol{x}} F(\boldsymbol{x}) > -\infty$*. For any* $\varepsilon \in (0, 1)$*, the sequence* $\{\boldsymbol{x}_t\}$ *obtained from update rule (13) satisfies*

$$\min_{j=0:T-1} \|\nabla F(\boldsymbol{x}_j)\|_2^2 \leq \varepsilon,$$

*after* $T$ *steps, where*

$$T = \begin{cases} 1 + \left\lceil \frac{2(\beta_0 + 2(F(\boldsymbol{x}_0) - F^*)/\alpha)(F(\boldsymbol{x}_0) - F^*)}{\varepsilon} \right\rceil & \text{if } \beta_0 \geq L \text{ or } \beta_1 \geq L, \\ 1 + \left\lceil \frac{\log(\frac{L}{\beta_0})}{\log(\frac{\varepsilon}{lL^2}+1)} \right\rceil + \left\lceil \frac{\left(L + \left(1+\frac{2}{\alpha}\right)\left(F(\boldsymbol{x}_0) - F^* + \frac{lL(L-\beta_0)}{2\beta_0}\right)\right)^2}{\varepsilon} \right\rceil & \text{otherwise.} \end{cases}$$

**Theorem 22** (Convergence rate of update rule (14)). *Suppose that* $\varphi \in C_l^{1,1}([1, +\infty))$*,* $\varphi$ *is* $\alpha$*-strongly convex, and* $F \in C_L^{1,1}(\mathbb{R}^d)$*,* $F^* = \inf_{\boldsymbol{x}} F(\boldsymbol{x}) > -\infty$*. For any* $\varepsilon \in (0, 1)$*, the sequence* $\{\boldsymbol{x}_t\}$ *obtained from update rule (14) satisfies*

$$\min_{j=0:T-1} \|\nabla F(\boldsymbol{x}_j)\|_2^2 \leq \varepsilon$$

*after* $T$ *steps, where*

$$T = \begin{cases} 1 + \left\lceil \frac{2(\beta_0 + \|\boldsymbol{g}_0\|_2^2/(\alpha\beta_0) + 2(F(\boldsymbol{x}_0) - F^*)/\alpha)(F(\boldsymbol{x}_0) - F^*)}{\varepsilon} \right\rceil & \text{if } \beta_0 \geq L \text{ or } \beta_1 \geq L, \\ 1 + \left\lceil \frac{\log(\frac{L}{\beta_0})}{\log(\frac{\varepsilon}{lL^2}+1)} \right\rceil + \left\lceil \frac{\left(L + \frac{2l}{\alpha\beta_0}L^2 + \frac{2l}{\alpha}L + \left(1+\frac{8}{\alpha}\right)\left(F(\boldsymbol{x}_0) - F^* + \frac{lL(L-\beta_0)}{2\beta_0}\right)\right)^2}{\varepsilon} \right\rceil & \text{otherwise.} \end{cases}$$

**Theorem 23** (Convergence rate of update rule (33)). *Suppose that* $\varphi \in C_l^{1,1}([1, +\infty))$*,* $\varphi$ *is* $\alpha$*-strongly convex, and* $F \in C_L^{1,1}(\mathbb{R}^d)$*,* $F^* = \inf_{\boldsymbol{x}} F(\boldsymbol{x}) > -\infty$*. For any* $\varepsilon \in (0, 1)$*, the sequence* $\{\boldsymbol{x}_t\}$ *obtained from update rule (33) satisfies*

$$\min_{j=0:T-1} \|\nabla F(\boldsymbol{x}_j)\|_2^2 \leq \varepsilon$$

*after* $T$ *steps, where*

$$T = \begin{cases} 1 + \left\lceil \frac{2(\beta_0 + 2(F(\boldsymbol{x}_0) - F^*)/\alpha)(F(\boldsymbol{x}_0) - F^*)}{\varepsilon} \right\rceil & \text{if } \beta_0 \geq L, \\ 2 + \left\lceil \frac{\left(\beta_0 + \frac{\|\boldsymbol{g}_0\|_2^2}{\alpha\beta_0} + \left(1+\frac{2}{\alpha}\right)\left(F(\boldsymbol{x}_0) - F^* + \frac{lL\|\boldsymbol{g}_0\|_2^2}{2\alpha\beta_0^2}\right)\right)^2}{\varepsilon} \right\rceil & \text{if } \beta_0 < L, \beta_1 \geq L, \\ 1 + \left\lceil \frac{\log(\frac{L}{\beta_0})}{\log(\frac{\varepsilon}{lL^2}+1)} \right\rceil + \left\lceil \frac{\left(\left(1+\frac{2l}{\alpha}\right)L + \frac{2lL^2}{\alpha\beta_0} + \left(1+\frac{2}{\alpha}\right)\left(F(\boldsymbol{x}_0) - F^* + \frac{lL}{2\beta_0}\left(\left(1+\frac{2l}{\alpha}\right)L + \frac{2lL^2}{\alpha\beta_0} - \beta_0\right)\right)\right)^2}{\varepsilon} \right\rceil & \text{otherwise.} \end{cases}$$

We begin our proof by following lemma.

**Lemma 24.** *Suppose $\varphi \in C_l^{1,1}(\mathbb{R}_{++})$. Fix $\varepsilon \in (0,1]$. In both update rules (13) and (14), after*
$$T = \left\lceil \frac{\log(\frac{L}{\beta_0})}{\log(\frac{\varepsilon}{lL^2}+1)} \right\rceil + 1 \text{ steps, either } \min_{t=0:T-1} \|\boldsymbol{g}_t\|_2^2 \leq \varepsilon, \text{ or } \beta_T \geq L \text{ holds.}$$

*Proof.* Assume that $\beta_T < L$ and $\min_{t=0:T-1} \|\boldsymbol{g}_t\|_2^2 > \varepsilon$. Recall that the sequence $\{\beta_t\}$ is an increasing sequence. Hence, $\beta_t < L$ for $0 \leq t \leq T$.
So, for all $0 \leq t \leq T-1$,

$$\varphi'\left(\frac{\beta_{t+1}}{\beta_t}\right) = \frac{\|\boldsymbol{g}_t\|_2^2}{\beta_{t+1}^2} > \frac{\varepsilon}{L^2} \text{ (in Algorithm 1),}$$

$$\varphi'\left(\frac{\beta_{t+1}}{\beta_t}\right) = \frac{\|\boldsymbol{g}_t\|_2^2}{\beta_t^2} > \frac{\varepsilon}{L^2} \text{ (in Algorithm 2).}$$

Note that $\varphi$ is a $l$-smooth convex function, and $\beta_{t+1}/\beta_t \geq 1$. So

$$\varphi'\left(\frac{\beta_{t+1}}{\beta_t}\right) \leq l\left(\frac{\beta_{t+1}}{\beta_t} - 1\right), \tag{34}$$

then

$$\frac{\beta_{t+1}}{\beta_t} > \frac{\varepsilon}{lL^2} + 1.$$

In this case,

$$L > \beta_T = \beta_0 \left(\frac{\varepsilon}{lL^2} + 1\right)^T$$

holds but it is impossible according to the setting of $T$ in the lemma. $\square$

We first prove Theorem 21 using following lemma.

**Lemma 25.** *In update rule (13), suppose $F \in C_L^{1,1}(\mathbb{R}^d)$, $\varphi \in C_l^{1,1}(\mathbb{R}_{++})$, and $\varphi$ is $\alpha$-strongly convex function. Denote $F^* = \inf_{\boldsymbol{x}} F(\boldsymbol{x}) > -\infty$. Let $t_0 \geq 1$ be the first index such that $\beta_{t_0} \geq L$. Then for all $t \geq t_0$,*

$$\beta_t \leq \beta_{t_0-1} + \frac{2}{\alpha}(F(\boldsymbol{x}_{t_0-1}) - F^*), \tag{35}$$

*and moreover,*

$$F(\boldsymbol{x}_{t_0-1}) - F^* \leq F(\boldsymbol{x}_0) - F^* + \frac{Ll}{2\beta_0}(\beta_{t_0-1} - \beta_0) \tag{36}$$

*Proof.* Same as equation (32),

$$F(\boldsymbol{x}_{t+1}) \leq F(\boldsymbol{x}_t) - \frac{1}{\beta_{t+1}}\left(1 - \frac{L}{2\beta_{t+1}}\right)\|\boldsymbol{g}_t\|_2^2.$$

For $t \geq t_0 - 1$, $\beta_{t+1} \geq L$, so

$$F(\boldsymbol{x}_{t+1}) \leq F(\boldsymbol{x}_t) - \frac{1}{2\beta_{t+1}}\|\boldsymbol{g}_t\|_2^2.$$

Hence, for all $k \geq 0$,

$$F(\boldsymbol{x}_{t_0+k}) \leq F(\boldsymbol{x}_{t_0-1}) - \frac{1}{2}\sum_{i=0}^{k}\frac{\|\boldsymbol{g}_{t_0+i-1}\|_2^2}{\beta_{t_0+i}}, \tag{37}$$

i.e.,

$$\sum_{i=0}^{k}\frac{\|\boldsymbol{g}_{t_0+i-1}\|_2^2}{\beta_{t_0+i}} \leq 2(F(\boldsymbol{x}_{t_0-1}) - F^*). \tag{38}$$

Note that $\varphi$ is $\alpha$-strongly convex and $\beta_{t+1}^2 \varphi'(\beta_{t+1}/\beta_t) = \|\boldsymbol{g}_t\|_2^2$. So

$$\frac{\|\boldsymbol{g}_t\|_2^2}{\beta_{t+1}} = \beta_{t+1}\varphi'\left(\frac{\beta_{t+1}}{\beta_t}\right) \geq \alpha\beta_t\left(\frac{\beta_{t+1}}{\beta_t} - 1\right),$$

and

$$\beta_{t+1} - \beta_t \leq \frac{1}{\alpha}\frac{\|\boldsymbol{g}_t\|_2^2}{\beta_{t+1}}. \tag{39}$$

Combining equation (38) and equation (39), we have

$$\beta_{t_0+k} \leq \beta_{t_0-1} + \frac{1}{\alpha}\sum_{i=0}^{k}\frac{\|\boldsymbol{g}_{t_0+i-1}\|_2^2}{\beta_{t_0+i}}$$

$$\leq \beta_{t_0-1} + \frac{2}{\alpha}(F(\boldsymbol{x}_{t_0-1}) - F^*).$$

We remain to give an a upper bound for $F(\boldsymbol{x}_{t_0-1})$ in the case $t_0 > 1$. Using equation (32) again, we get

$$F(\boldsymbol{x}_{t_0-1}) - F(\boldsymbol{x}_0) \leq \sum_{i=0}^{t_0-2} -\frac{1}{\beta_{i+1}}\left(1 - \frac{L}{2\beta_{i+1}}\right)\|\boldsymbol{g}_i\|_2^2 \leq \frac{L}{2}\sum_{i=0}^{t_0-2}\frac{\|\boldsymbol{g}_i\|_2^2}{\beta_{i+1}^2}$$

$$= \frac{L}{2}\sum_{i=0}^{t_0-2}\varphi'\left(\frac{\beta_{i+1}}{\beta_i}\right) \leq \frac{Ll}{2}\sum_{i=0}^{t_0-2}\left(\frac{\beta_{i+1}}{\beta_i} - 1\right)$$

$$\leq \frac{Ll}{2}\sum_{i=0}^{t_0-2}\left(\frac{\beta_{i+1} - \beta_i}{\beta_0}\right) = \frac{Ll}{2\beta_0}(\beta_{t_0-1} - \beta_0).$$

In the above, the second inequality follows from the assumed $l$-smoothness of $\varphi$, and the last inequality follows from $\beta_t \geq \beta_0$ for all $t \geq 0$. $\qquad\square$

***proof of Theorem 21.*** If $t_0 = 1$, by equation (37), for all $t \geq 1$, we have

$$F(\boldsymbol{x}_t) \leq F(\boldsymbol{x}_0) - \frac{1}{2}\sum_{i=0}^{t-1}\frac{\|\boldsymbol{g}_i\|_2^2}{\beta_{i+1}}$$

$$\leq F(\boldsymbol{x}_0) - \frac{1}{2}\sum_{i=0}^{t-1}\frac{\|\boldsymbol{g}_i\|_2^2}{\beta_0 + \frac{2}{\alpha}(F(\boldsymbol{x}_0) - F^*)}.$$

Then after $T = 1 + \left\lceil\frac{2(\beta_0 + 2(F(\boldsymbol{x}_0) - F^*)/\alpha)(F(\boldsymbol{x}_0) - F^*)}{\varepsilon}\right\rceil$ steps,

$$\min_{t=0:T-1}\|\boldsymbol{g}_t\|_2^2 \leq \frac{1}{T}\sum_{t=0}^{T-1}\|\boldsymbol{g}_t\|_2^2$$

$$\leq \frac{2}{T}(F(\boldsymbol{x}_0) - F^*)(\beta_0 + \frac{2}{\alpha}(F(\boldsymbol{x}_0) - F^*)) \leq \varepsilon.$$

Otherwise, if $t_0 > 1$, we have $\beta_{t_0-1} < L$. Then for all $t \geq t_0$,

$$\beta_t \leq L + \frac{2}{\alpha}\left(F(\boldsymbol{x}_0) - F^* + \frac{lL(L - \beta_0)}{2\beta_0}\right) \tag{40}$$

Denote the right hand of equation (40) as $\beta_{max}$. Using equation (37) again, for we have

$$F(\boldsymbol{x}_{t_0+M}) \leq F(\boldsymbol{x}_{t_0-1}) - \frac{1}{2}\sum_{i=0}^{M}\frac{\|\boldsymbol{g}_{t_0+i-1}\|_2^2}{\beta_{t_0+i}}$$

$$\leq F(\boldsymbol{x}_{t_0-1}) - \frac{1}{2\beta_{max}}\sum_{i=0}^{M}\|\boldsymbol{g}_{t_0+i-1}\|_2^2.$$

Hence,

$$
\begin{aligned}
\min_{t=0:t_0+M-1} \|\boldsymbol{g}_t\|_2^2 &\leq \min_{t=t_0-1:t_0+M-1} \|\boldsymbol{g}_t\|_2^2 \\
&\leq \frac{1}{M+1} \sum_{i=0}^{M} \|\boldsymbol{g}_{t_0+i-1}\|_2^2 \\
&\leq \frac{1}{M+1} 2\beta_{max}(F(\boldsymbol{x}_{t_0-1}) - F^*) \\
&\leq \frac{2\beta_{max}}{M+1} \left( F(\boldsymbol{x}_0) - F^* + \frac{lL(L-\beta_0)}{2\beta_0} \right).
\end{aligned}
$$

At last, with recalling the conclusion of Lemma 24, after

$$
T = \left\lceil \frac{\log(\frac{L}{\beta_0})}{\log(\frac{\varepsilon}{lL^2}+1)} \right\rceil + \left\lceil \frac{2\beta_{max}}{\varepsilon} \left( F(\boldsymbol{x}_0) - F^* + \frac{lL(L-\beta_0)}{2\beta_0} \right) \right\rceil + 1
$$

steps, we have $\min_{t=0:T-1} \|\boldsymbol{g}_t\|_2^2 \leq \varepsilon$. $\qquad\square$

Next we prove Theorem 22.

**Lemma 26.** *In update rule (14), suppose $F \in C_L^{1,1}(\mathbb{R}^d)$, $\varphi \in C_l^{1,1}(\mathbb{R}_{++})$, and $\varphi$ is $\alpha$-strongly convex function. Denote $F^* = \inf_{\boldsymbol{x}} F(\boldsymbol{x})$. Let $t_0 \geq 1$ be the first index such that $\beta_{t_0} \geq L$. Then for all $t \geq t_0$,*

$$
\beta_t \leq \beta_{t_0} + \frac{8}{\alpha}(F(\boldsymbol{x}_{t_0-1}) - F^*), \tag{41}
$$

*and moreover,*

$$
F(\boldsymbol{x}_{t_0-1}) - F^* \leq F(\boldsymbol{x}_0) - F^* + \frac{Ll}{2\beta_0}(\beta_{t_0-1} - \beta_0), \tag{42}
$$

$$
\beta_{t_0} \leq
\begin{cases}
\beta_0 + \frac{\|\boldsymbol{g}_0\|_2^2}{\alpha\beta_0} & \text{if } t_0 = 1, \\
L + \frac{2l}{\alpha\beta_0}L^2 + \frac{2l}{\alpha}L & \text{if } t_0 \geq 2,
\end{cases} \tag{43}
$$

*Proof.* Same as the proof of Lemma 25, we first get for all $k \geq 0$,

$$
\sum_{i=0}^{k} \frac{\|\boldsymbol{g}_{t_0+i-1}\|_2^2}{\beta_{t_0+i}} \leq 2(F(\boldsymbol{x}_{t_0-1}) - F^*).
$$

Note that in update rule (14), $\beta_t^2 \varphi'(\beta_{t+1}/\beta_t) = \|\boldsymbol{g}_t\|_2^2$. So

$$
\begin{aligned}
\beta_{t_0+k+1} &= \beta_{t_0+k} + \beta_{t_0+k} \left( \frac{\beta_{t_0+k+1}}{\beta_{t_0+k}} - 1 \right) \\
&\leq \beta_{t_0+k} + \frac{\beta_{t_0+k}}{\alpha} \varphi' \left( \frac{\beta_{t_0+k+1}}{\beta_{t_0+k}} \right) = \beta_{t_0+k} + \frac{1}{\alpha} \frac{\|\boldsymbol{g}_{t_0+k}\|_2^2}{\beta_{t_0+k}} \\
&\leq \beta_{t_0+k} + \frac{2}{\alpha} \frac{\|\boldsymbol{g}_{t_0+k} - \boldsymbol{g}_{t_0+k-1}\|_2^2 + \|\boldsymbol{g}_{t_0+k-1}\|_2^2}{\beta_{t_0+k}} \\
&\leq \beta_{t_0+k} + \frac{2}{\alpha} \frac{L^2 \|\boldsymbol{x}_{t_0+k} - \boldsymbol{x}_{t_0+k-1}\|_2^2 + \|\boldsymbol{g}_{t_0+k-1}\|_2^2}{\beta_{t_0+k}} \\
&\leq \beta_{t_0+k} + \frac{2}{\alpha} \frac{L^2 \|\boldsymbol{g}_{t_0+k-1}\|_2^2}{\beta_{t_0+k}^3} + \frac{2}{\alpha} \frac{\|\boldsymbol{g}_{t_0+k-1}\|_2^2}{\beta_{t_0+k}} \\
&\leq \beta_{t_0+k} + \frac{4}{\alpha} \frac{\|\boldsymbol{g}_{t_0+k-1}\|_2^2}{\beta_{t_0+k}} \leq \beta_{t_0} + \frac{4}{\alpha} \sum_{i=0}^{k} \frac{\|\boldsymbol{g}_{t_0+i-1}\|_2^2}{\beta_{t_0+i}} \\
&\leq \beta_{t_0} + \frac{8}{\alpha}(F(\boldsymbol{x}_{t_0-1}) - F^*).
\end{aligned}
$$

If $t_0 = 1$, then

$$\beta_{t_0} \leq \beta_0 + \frac{\|\boldsymbol{g}_0\|_2^2}{\alpha \beta_0},$$

and if $t_0 \geq 2$, then

$$
\begin{aligned}
\beta_{t_0} &\leq \beta_{t_0-1} + \frac{\|\boldsymbol{g}_{t_0-1}\|_2^2}{\alpha \beta_{t_0-1}} \\
&= \beta_{t_0-1} + \frac{2L^2}{\alpha} \frac{\|\boldsymbol{g}_{t_0-2}\|_2^2}{\beta_{t_0-1}^3} + \frac{2}{\alpha} \frac{\|\boldsymbol{g}_{t_0-2}\|_2^2}{\beta_{t_0-2}} \\
&\leq \beta_{t_0-1} + \frac{2L^2}{\alpha} \frac{l(\beta_{t_0-1} - \beta_{t_0-2})\beta_{t_0-2}}{\beta_{t_0-1}^3} + \frac{2}{\alpha} l(\beta_{t_0-1} - \beta_{t_0-2}) \\
&\leq L + \frac{2l}{\alpha \beta_0} L^2 + \frac{2l}{\alpha} L.
\end{aligned}
$$

At last, for $t_0 > 0$, we have

$$
\begin{aligned}
F(\boldsymbol{x}_{t_0-1}) - F(\boldsymbol{x}_0) &\leq \sum_{i=0}^{t_0-2} -\frac{1}{\beta_{i+1}} \left(1 - \frac{L}{2\beta_{i+1}}\right) \|\boldsymbol{g}_i\|_2^2 \\
&\leq \frac{L}{2} \sum_{i=0}^{t_0-2} \frac{\|\boldsymbol{g}_i\|_2^2}{\beta_{i+1}^2} \leq \frac{L}{2} \sum_{i=0}^{t_0-2} \frac{\|\boldsymbol{g}_i\|_2^2}{\beta_i^2} \\
&= \frac{L}{2} \sum_{i=0}^{t_0-2} \varphi'\left(\frac{\beta_{i+1}}{\beta_i}\right) \leq \frac{Ll}{2} \sum_{i=0}^{t_0-2} \left(\frac{\beta_{i+1}}{\beta_i} - 1\right) \\
&\leq \frac{Ll}{2} \sum_{i=0}^{t_0-2} \left(\frac{\beta_{i+1} - \beta_i}{\beta_0}\right) = \frac{Ll}{2\beta_0}(\beta_{t_0-1} - \beta_0).
\end{aligned}
$$

$\square$

***proof of Theorem 22***. The proof is completely similar to the proof of Theorem 21. $\square$

Next, we prove Theorem 23.

**Lemma 27.** *In update rule (33), suppose $F \in C_L^{1,1}(\mathbb{R}^d)$, $\varphi \in C_l^{1,1}(\mathbb{R}_{++})$, and $\varphi$ is $\alpha$-strongly convex function. Denote $F^* = \inf_{\boldsymbol{x}} F(\boldsymbol{x})$. Let $t_0 \geq 0$ be the first index such that $\beta_{t_0} \geq L$. Then for all $t \geq t_0$,*

$$\beta_t \leq \beta_{t_0} + \frac{2}{\alpha}(F(\boldsymbol{x}_{t_0}) - F^*), \tag{44}$$

*and moreover,*

$$F(\boldsymbol{x}_{t_0}) - F^* \leq F(\boldsymbol{x}_0) - F^* + \frac{Ll}{2\beta_0}(\beta_{t_0} - \beta_0), \tag{45}$$

$$\beta_{t_0} \leq \begin{cases} \beta_0 + \frac{\|\boldsymbol{g}_0\|_2^2}{\alpha \beta_0}, & \text{if } t_0 \geq 1, \\ \left(1 + \frac{2l}{\alpha}\right) L + \frac{2l}{\alpha \beta_0} L^2, & \text{if } t_0 \geq 2. \end{cases} \tag{46}$$

*Proof.* Same as the proof of Lemma 25, we first get

$$
\begin{aligned}
F(\boldsymbol{x}_{t_0+k+1}) &\leq F(\boldsymbol{x}_{t_0+k}) - \frac{1}{\beta_{t_0+k}} \left(1 - \frac{L}{2\beta_{t_0+k}}\right) \|\boldsymbol{g}_{t_0+k}\|_2^2 \\
&\leq F(\boldsymbol{x}_{t_0+k}) - \frac{1}{2\beta_{t_0+k}} \|\boldsymbol{g}_{t_0+k}\|_2^2 \\
&\leq F(\boldsymbol{x}_{t_0}) - \frac{1}{2} \sum_{i=0}^{k} \frac{\|\boldsymbol{g}_{t_0+i}\|_2^2}{\beta_{t_0+i}},
\end{aligned}
$$

and

$$\sum_{i=0}^{k} \frac{\|\boldsymbol{g}_{t_0+i}\|_2^2}{\beta_{t_0+i}} \le 2(F(\boldsymbol{x}_{t_0}) - F^*).$$

Note that in Algorithm 2, $\beta_t^2 \varphi'(\beta_{t+1}/\beta_t) = \|\boldsymbol{g}_t\|_2^2$. So

$$\begin{aligned}
\beta_{t_0+k+1} &= \beta_{t_0+k} + \beta_{t_0+k}\left(\frac{\beta_{t_0+k+1}}{\beta_{t_0+k}} - 1\right) \\
&\le \beta_{t_0+k} + \frac{\beta_{t_0+k}}{\alpha}\varphi'\left(\frac{\beta_{t_0+k+1}}{\beta_{t_0+k}}\right) \\
&= \beta_{t_0+k} + \frac{1}{\alpha}\frac{\|\boldsymbol{g}_{t_0+k}\|_2^2}{\beta_{t_0+k}} \\
&\le \beta_{t_0} + \frac{1}{\alpha}\sum_{i=0}^{k}\frac{\|\boldsymbol{g}_{t_0+i}\|_2^2}{\beta_{t_0+i}} \\
&\le \beta_{t_0} + \frac{2}{\alpha}(F(\boldsymbol{x}_{t_0}) - F^*).
\end{aligned}$$

At last, for $t_0 > 0$, we have

$$\begin{aligned}
F(\boldsymbol{x}_{t_0}) - F(\boldsymbol{x}_0) &\le \frac{L}{2}\sum_{i=0}^{t_0-1}\frac{\|\boldsymbol{g}_i\|_2^2}{\beta_i^2} \\
&= \frac{L}{2}\sum_{i=0}^{t_0-1}\varphi'\left(\frac{\beta_{i+1}}{\beta_i}\right) \\
&\le \frac{Ll}{2\beta_0}(\beta_{t_0} - \beta_0).
\end{aligned}$$

Recall that

$$\alpha\beta_t(\beta_{t+1} - \beta_t) \le \|\boldsymbol{g}_t\|_2^2 \le l\beta_t(\beta_{t+1} - \beta_t).$$

So if $t_0 = 1$,

$$\beta_{t_0} \le \beta_0 + \frac{\|\boldsymbol{g}_0\|_2^2}{\alpha\beta_0}.$$

And if $t_0 \ge 2$, we have

$$\begin{aligned}
\beta_{t_0} &\le \beta_{t_0-1} + \frac{\|\boldsymbol{g}_{t_0-1}\|_2^2}{\alpha\beta_{t_0-1}} \\
&\le \beta_{t_0-1} + 2\frac{\|\boldsymbol{g}_{t_0-1} - \boldsymbol{g}_{t_0-2}\|_2^2 + \|\boldsymbol{g}_{t_0-2}\|_2^2}{\alpha\beta_{t_0-1}} \\
&\le \beta_{t_0-1} + \frac{2L^2}{\alpha}\frac{\|\boldsymbol{x}_{t_0-1} - \boldsymbol{x}_{t_0-2}\|_2^2}{\beta_{t_0-1}} + \frac{2}{\alpha}\frac{\|\boldsymbol{g}_{t_0-2}\|_2^2}{\beta_{t_0-2}} \\
&= \beta_{t_0-1} + \frac{2L^2}{\alpha}\frac{\|\boldsymbol{g}_{t_0-2}\|_2^2}{\beta_{t_0-2}^2\beta_{t_0-1}} + \frac{2}{\alpha}\frac{\|\boldsymbol{g}_{t_0-2}\|_2^2}{\beta_{t_0-2}} \\
&\le \beta_{t_0-1} + \frac{2L^2}{\alpha}\frac{l(\beta_{t_0-1} - \beta_{t_0-2})}{\beta_{t_0-2}\beta_{t_0-1}} + \frac{2}{\alpha}l(\beta_{t_0-1} - \beta_{t_0-2}) \\
&\le L + \frac{2l}{\alpha\beta_0}L^2 + \frac{2l}{\alpha}L.
\end{aligned}$$

$\square$

***proof of Theorem 23***. The proof is completely similar to the proof of Theorem 21. $\qquad\square$

