# OpenReview forum: "Hyper-Regularization: An Adaptive Choice for the Learning Rate in Gradient Descent"
_ICLR.cc/2019/Conference_

### Official Review · AnonReviewer1 · 2018-10-26
**Unclear formulation, and Benefit of approach is not Demonstrated**

**Rating:** 4
**Confidence:** 4

**Review:**

 Summary:
%%%%%%%%%%%%%%%
The paper explores ways to adapt the learning rate rule through a new minimax formulation.
The authors provide regret bounds for their method in the online convex optimization setting.

Comments:
%%%%%%%%%%%%%%%
-I found the motivation of the approach to be very lacking.
Concretely, it is not clear at all why the minimax formulation even makes sense, and the authors do not explain this issue.

-While the authors provide regret guarantees for their method, the theoretical analysis does not reflect when is their approach  beneficial compared to standard adaptive methods. Concretely, their bounds compare with the well known bounds of AdaGrad.
It is nice that their approach enables to extract AdaGrad as a private case. But again, it is not clear what is the benefit of their extension.

-Finally, the experiments do not illustrate almost any benefit of the new approach compared to standard adaptive methods.


Summary
%%%%%%%%%%%%%%%
The paper suggests a different approach to adapt the learning rate.
Unfortunately, the reasoning behind the new approach is not very clear.
Also, nor theory neither experiments illustrate the benefit of this new approach over standard methods.

---

> ### Author Response · Authors · 2018-11-23
> **Thanks for your time and insightful comments!**
>
> 1) Our main contribution (or focus) of this paper is to propose a framework for adjusting learning rate adaptively. We offer a novel viewpoint different from previous main approaches like line search and approximate second-order methods (BBstep, Adagrad, etc.).
>
> 2) Our idea stems  from  the  work  of  Daubechies  et  al.  (2010),  where  the  authors  adjusted  the weights of the weighted least squares problem by solving an extra objective function which added a regularizer about the weights to origin objective function.
>
> 3) Since our framework can derive AdaGrad as a private case, we are more general scene so our bound doesn’t better than AdaGrad. However, our bound are almost same as AdaGrad.

---

> > ### Comment · AnonReviewer1 · 2018-12-03
> > **Comment**
> >
> > I thank the reviewers for their response, and I keep my score.

---

### Official Review · AnonReviewer2 · 2018-10-28
**Novel idea but theoretical guarantees and empirical results are not convincing.**

**Rating:** 4
**Confidence:** 4

**Review:**

This paper presents a method for adaptively tuning the learning rate in gradient descent methods. The authors consider the formulation of each gradient descent update as a quadratic minimization problem and they propose adding a phi-divergence between the learning rate that would be used and an auxiliary vector. The authors also propose adding a maximization over all learning rates in the update.

The authors study an important problem and propose a novel method. The algorithms suggested by the author are also relatively clear, and it is great that the paper presents both theoretical results as well as numerical experiments.

On the other hand, I didn't find the main idea of hyper-regularization to be well-justified. It is not clear why adding an additional regularization term for the learning rate makes sense , and it is even less clear why this should be presented as a maxmin problem. This can make the update step much more complicated and is probably why the authors also propose a simpler alternating optimization algorithm as an alternative. Unfortunately, the authors do not discuss how this alternating optimization problem relates to the original one, and the theoretical guarantees are only presented for the original algorithm. The authors also do not justify the choice of phi-divergence as the regularizer for the learning rate. The theoretical guarantees in the paper also do not suggest that the algorithm presented in the paper is better than existing state-of-the-art methods, even in specific situations (i.e. the regret bounds don't appear better than the AdaGrad regret bounds). Moreover, without tests for statistical significance, I also didn't find the experimental results sufficiently compelling.

Specific comments and questions:
1) Page 3: Equation (4): The paper would be stronger if the authors motivated why the regularization should be posed as an outer maximization.
2) Page 3: "we use the \phi-divergence as our hyper-regularization". Why is this a good choice of reuglarizer?
3) Page 3: "only a few extra calculations are required for each step". This is a misleading comment, because the maximization can be hard when phi is complicated, even if the problem splits across dimensions.
4) Page 4: "The solution of problem (5) is the same as (7) in unconstrained case". You should provide a reference for this statement as well as discuss the specific assumptions on the objective that allow you to arrive at this claim.
5) Page 4: "while the solution of (7) is more difficult to get. Thus, we choose (5) as our basic problem". This seems like a very bad motivation for choosing the maxmin formulation. For instance, the problem would be even simpler if  you didn't include this extra phi-divergence at all.
6) Page 4: "Although setting \eta-t=\beta_t is our main focus...". Why is smoothness in the learning rate a good property?
7) Page 5: Equation (11). How do these iterates relate to the ones in equation (5) (e.g. when do they coincide, if ever)?
8) Page 5: "influence the efficient of our algorithms." Grammatical error.
9) Page 6. "our algorithms are robust to the choice of initial learning rates and do not rely on the Lipschitz constant or smoothness constant". I'm not sure why this is a valuable property, since AdaGrad doesn't rely on these parameters either.
10) Page 6: Theorems 6 and 7. How do these results depend on alpha and \beta_t? This paper would be much stronger if the bounds depend on \phi more clearly and if the authors were able to show that there exist choices of phi that make this algorithm better than existing methods.
11) Page 6: Theorem 7: The dependence on G in the regret bound actually makes this worse than the AdaGrad regret bound.
12) Page 7: "KL_devergence". Typo.
13) Page 7: "different update rules were compared in advance to select the specific one for any phi divergence in the following experiments." What does this mean exactly? How much of a difference does the choice of update rule make?
14) Page 7: "growth clipping is applied to all algorithms in our framework". Why is this necessary, and how does it affect the theoretical results?
14) Page 7-8: Figures 1, 2, and 3. It's hard to interpret the significance of these results without error bars.

---

> ### Author Response · Authors · 2018-11-23
> **Thanks for your time and helpful comments!**
>
> 1) Our idea stems  from  the  work  of  Daubechies  et  al.  (2010),  where  the  authors  adjusted  the weights of the weighted least squares problem by solving an extra objective function which added a regularizer about the weights to origin objective function.
> 2) We give theoretical analysis for both update rules (see Theorems 6 and 7) not only for the original algorithm.
> 3) Compared with Bregman divergence, \phi-divergence naturally imposes nonnegativity on \beta and \veta_t, which is necessary for learning rates.
>
>
> Specific comments and questions:
>
> 1) Since taking a convex regularization term of learning rate and viewing it as a penalty according to current learning rate, we need to regard this as a maximum problem. And there is little difference when adding a regularization and viewing it as a minimum problem. Actually, our paper proposes a framework of adaptive step size learning.
>
> 2)Since \phi-divergence is natural for nonnegative variable, compared with Bregman divergence, and doesn’t have to be a probability (sum is 1) compared to KL-divergence.
>
> 3) We have shown some common \phi-divergence and relevant update rules in Appendix D.1, and most of them seem simple to solve.
>
> 4) The objective functions of  problems (5) and (7) are concave for \beta and convex for x (since \beta > 0), and let the equations of both partial derivatives equal zero (it is somehow like the equation(12) ), then the solutions satisfy saddle point condition, so the solutions are identical.
>
> 5) We have shown in Lemma 2 that the solution of (5) can easily be extended to constrained cases. Moreover, in practical use, growth clipping is often necessary, which indicates constrained cases. Therefore, we choose maxmin formulation instead of minmax formulation.
>
> 6) We would like to adaptively choose the learning rate in the optimization period rather than setting priori \eta_t for \beta_t, while not on account of the smoothness.
>
> 7) Equation (11), got by alternating update rule, first uses recommended step size \eta to minimizes x, and then maximizes \beta to get next step size, next turn back to get x by using this step size. Our theorem showed the convergence bound for both methods. Perhaps, there is no need to show when they coincide.
>
> 9) First, we focus on the convergence with different choice of initial learning rate, while many methods, like GD, would fail for extremely large initial learning rate. However, this doesn't bother us at all since we are free of choosing the initial rate. Second, like GD and many other optimization methods, the choice of initial learning rate may need Lipschitz constant or smoothness constant for the sake of convergence, but in our methods, it doesn't.
>
> 10) We propose a novel framework on adaptively updating the learning rate with a regularization term, which we take \phi divergence as an example in the paper. In this way, the bounds are given for \phi divergence generally. Therefore, we do not make special assumptions on \alpha or \beta_t.
>
> 11) The bounds are given for general \phi divergence. Viewed as a special case of our framework with a specialized \phi divergence, AdaGrad is of no wonder to enjoy lower regret bound than the general regret bound we give.
>
> 13) Due to the space limitation, our description may mislead you. Given a \phi divergence, we totally have three different ways on how to update x_t and \beta_t alternately, corresponding to Algorithms 1 and 2 in page 5, and Algorithm 3 in Appendix C. Actually, we should not mention Algorithm 1 here, for it is shown in Appendix D.1 that Algorithm 1 is always computationally unfriendly. As shown in Figure 1, Algorithm 2 outperforms Algorithm 3 when the initial learning rate is extremely large. However, at their best initial learning rates, their performance is comparable. Out of the consideration on training stability with large initial learning rate, we finally choose the second update rule corresponding to Algorithm 2.
>
> 14) Growth clipping does not affect the theoretical result, since it is only applied to the algorithms in the experiment. The practical optimization problem neither generally guarantees the gradient smoothness, nor guarantees the global strong convexity. So it is necessary to apply growth clipping in case of training collapses due to the stochastic gradient.
>
> 15)Figure 2 describes the results of experiments carried out in the full gradient setting. Since there is no randomness, it is of no use to carry out duplicated experiments.

---

### Official Review · AnonReviewer3 · 2018-11-03
**minor generalization of AdaGrad style methods**

**Rating:** 4
**Confidence:** 5

**Review:**

The paper presents a generalization of the Adagrad type methods using a min-max formulation and then presents two alternate algorithms to solve this formulation.

It is unclear to me that much extra generalization has been achieved over the original AdaGrad paper. That paper simply presents the choice of hyperparameters as an optimal solution to a proximal primal dual formulation. The formulation presented here appears to be another form of the proximal mapping formulation, and so it is unclear what the advance here is. The AdaGrad paper used a particular Bregman divergence, and different divergences yield slightly different methods, as is observed here by the authors when they use different divergence measures.

The Bregman divergences do make sense from a primal pual proximal formulation point of view, but why do you use a discrepancy function in your min-max formulation that comes from the \phi - divergence family? Why not consider an L_p normalization of the discrepancy?

The difference between formulations (5) and (6) is not clearly specified. Did you mean to drop the constraints that \beta \in \cal{B}_t ? Otherwise, why is (6) , which looks to be a re-write of (5), unconstrained and hence separable?

The authors claim that the method is free of parameter choices, but the initial \beta_0 seems to be a crucial parameter here since it forms both a target and a lower bound for subsequent \beta_t's. How is this parameter chosen and what effect does it have on convergence? From the results (Figs in Sec 5), this choice does significantly impact the final test loss obtained.

I could not find a proof for Thm 6 in the appendix. Did I over look it or is there a typo?

---

> ### Author Response · Authors · 2018-11-23
> **Thanks for your thoughtful review and your time!**
>
> 1) Our main contribution (or focus) of this paper is to propose a framework for adjusting learning rate adaptively. We offer a novel viewpoint different from previous main approaches like line search and approximate second-order methods (BBstep, Adagrad, etc.).
> 2) Compared with Bregman divergence, \phi-divergence naturally imposes nonnegative constraint about \beta and \eta_t, which is necessary for learning rates, while the L_p normalization still can’t guarantee nonnegative condition for learning rate.
> 3) Equation (6) is equivalent to (5). We just want to rewrite (5) to a more clear scheme.
> 4) In the classical gradient descent algorithm formulated as x_{t+1} = x_t - g_t / \beta, for small \beta, more precisely for \beta < 2 / L, the algorithm has no guarantee for convergence. Our framework gives a upper bound for runtime (O(1 / \varepsilon)) or regret (O(\sqrt(T))) for arbitrary \beta_0. Moreover, like Adagrad or gradient descent, algorithms derived from our framework also can be suggested a best initial learning rate for optimization ( based on our regret bounds).
> 5) The proof of Theorem 6 can be found in the proofs of Theorems 21, 22, and 23.

---

### Public Comment · (anonymous) · 2018-12-13
**initial gradient problem**

Hello,

During the implementation of the 2d neural network on MNIST using the proposed algorithm, I got a problem. The initial value of gradient is big (since I put the random values in Ws and they are not probably close to the optimum) so when I use this new learning rate, it doesn't converge. I want to ask if there is an initial value on the learning rate in your algorithm in order to avoid that?
One solution: we can do a step of regular gradient descent and after that change the update rules for iterations > 1.

Thanks

---

### Meta-Review · Area_Chair1 · 2018-12-18
**Reject**

**Confidence:** 4
**Recommendation:** Reject

**Metareview:**

All three reviewers found that the motivation for the proposed method was lacking and recommend rejection. The AC thus recommends the authors to take these comments in consideration when revising their manuscript.